# HERMES: Heterogeneous Effects Representation with Matched Embeddings using Siamese Networks

**Rocco Zaccagnino**                                    *rzaccagnino@unisa.it*
*Department of Computer Science*
*University of Salerno*

**Gerardo Benevento**                                   *gbenevento@unisa.it*
*Department of Computer Science*
*University of Salerno*

**Delfina Malandrino**                                  *dmalandrino@unisa.it*
*Department of Computer Science*
*University of Salerno*

**Donatello Telesca**                                   *dtelesca@g.ucla.edu*
*Department of Biostatistics*
*University of California Los Angeles*

**Alessia Ture**                                        *ature@unisa.it*
*Department of Computer Science*
*University of Salerno*

**Reviewed on OpenReview:** *https://openreview.net/forum?id=G6F0fkBQEG*

## Abstract

We consider the problem of estimating heterogeneous treatment effects from observational data. Specifically, we are interested in the estimation of conditional average treatment effects (CATE) functions, i.e. functions mapping the effect of a binary treatment to the space of unit-level covariates. In the absence of a controlled randomized mechanism of treatment assignment, simple comparisons between treated and control populations can be potentially confounded by significant distributional differences in the covariate space. In this context, recent *representation learning* strategies aim to learn balanced latent representations in a new space where the treated and control distributions are more comparable, reducing variance. We introduce HERMES (Heterogeneous Effects Representation with Matched Embeddings using Siamese Networks), a novel framework that integrates self-supervised contrastive learning into causal representation learning. HERMES employs a Siamese architecture that dynamically pairs individuals based on similarity in estimated individual treatment effects (ITE), encouraging representations where proximity reflects treatment-response similarity rather than covariate similarity alone. Unlike representation learning approaches that rely only on covariates, HERMES injects the ITE into representation learning, improving accuracy under standard assumptions. Experiments on IHDP and JOBS benchmarks show that HERMES improves the expected Precision in MSE by 14–15% over baselines, without added inference cost.

## 1 Introduction

Estimating the heterogeneous causal effect of a binary intervention is a fundamental task in many policy and scientific domains. We focus on estimates of the conditional average treatment effect (CATE) as the main estimand of interest in personalized treatment regimes (Chen & Xie, 2023). In *precision medicine*, for

example, targeted cancer therapies may be effective only for patients with certain mutations. Administering treatment indiscriminately exposes patients to unnecessary toxicity and risks. Estimating the CATE allows us to identify who is most likely to benefit from intervention (Harvey et al., 2020). Similar considerations apply whenever decision making is likely to ensue in heterogeneous unit-specific effects. See, for example, Athey & Imbens (2017) and LaLonde (1986) for applications of CATE to *policy-making and welfare*, or Ascarza (2018) for applications to *marketing strategies.*

Carefully designed randomized controlled trials (RCT) are the gold standard for measuring causal effects. However, these studies are often hindered by numerous practical limitations (Munk et al., 2020; Klonoff, 2020), and many inquiries must rely on the information included in *observational data.* Estimating CATE functions from observational data must, however, contend with a fundamental challenge: *confounding.* This arises when covariates that influence treatment assignment are also associated with the outcome, making it difficult to isolate the causal effect of treatment.

A common framework for careful causal analysis in observational settings relies on the idea of potential outcomes Imbens & Rubin (2015), where every observational unit is endowed with two "potential outcomes" corresponding to observations obtained with and without treatment. Clearly, each realized sample only allows for the observation of one of these two outcomes. Therefore, the unobserved "*counterfactual*" outcome precludes direct observation of the individual treatment effect (ITE). While this simple idea highlights why this problem differs from standard supervised learning, whenever treatment assignment and outcomes can be assumed conditionally independent, given measurement of a sufficient set of features (*unconfoundedness*), the causal effect is indeed identifiable through *local average differences.*

Crucially, local averaging aims to closely emulate the internal validity of a randomized controlled trial, and produce "locally" accurate estimates of the individual treatment effect despite the inaccessibility of the counterfactual outcome. To achieve this, the discrepancy in feature distributions for the treated and control subject groups are expressed via a balancing score (Rosembaum & Rubin, 1983), which is then used to apply a correction to the estimate of conditional outcome functions (Stuart, 2010; Rotnitzky & Robins, 1995; Athey et al., 2017).

Several methods trained to separately learn a balancing score and conditional outcome functions can be adapted to obtain consistent estimators (Künzel et al., 2019). In contrast, the representation learning literature has focused on end-to-end learning of CATE functions with balanced latent representations of the feature space (Shalit et al., 2017). Within this framework, the embedding distributions of treated and control groups are as similar as possible, thus stabilizing estimation and reducing variance. Recent work has further clarified important aspects concerning the interpretation of balancing learned representations. In particular, representation balancing is not a substitute for the identifying assumptions required in observational causal inference, nor can it eliminate bias arising from unobserved confounding. Rather, when adjustment is justified on the basis of the observed covariates, representation balancing should be viewed as a finite-sample regularization device that improves empirical comparability, promotes overlap, and stabilizes counterfactual prediction. This interpretation is consistent with recent analyses of representation-induced confounding bias, orthogonal representation learning, and overlap-adaptive regularization for CATE estimation (Melnychuk et al., 2024; 2026a;b).

This is often achieved using Neural Networks as function learners with balanced representations of the feature space learned through integral probability metric (IPM) regularization (Shalit et al., 2017), or by framing the problem in a multi-task learning context with shared embeddings predicting both treatment assignment and outcomes (Shi et al., 2019; Tesei et al., 2023). This strategy is justifiable from a theoretical perspective in that the expected error in learning ITE is upper bounded by two components: the error in learning both the factual and counterfactual outcomes plus a term quantifying the dissimilarity of the treated and control distributions of the learned embeddings (Shalit et al., 2017). An example within this framework is `BCAUSS` (Balancing Covariates Automatically Using Self-Supervision) (Tesei et al., 2023), which relies on a multi-task architecture and a self-supervised objective to encourage treated and control individuals to have similar latent representations.

At the same time, useful causal representations need not be driven solely by treatment-assignment information. Prior work has shown that low-dimensional summaries may also retain outcome-relevant or more

generally deconfounding structure, as in the literature on prognostic scores and related causal representations (Hansen, 2008; Huang & Chan, 2017; Luo & Zhu, 2020; D'Amour & Franks, 2021; Clivio et al., 2026). Taken together, these perspectives suggest that an effective causal representation should do more than make treated and control groups globally comparable: it should also preserve the structure in the covariates that is informative for treatment effect heterogeneity.

These representation strategies, however, are not designed to maintain local coherence for individuals with similar effects, and micro-regions may therefore exist in the latent space where two nearly identical individuals are projected far apart if their ITEs differ. Thus, we hypothesize the possibility to build and train a model on the correlation between latent space and ITE, in order to create representations whose similarity reflects a similar ITE value. In this work, we propose the estimation of heterogeneous treatment effects via representation learning with matched embeddings using Siamese networks (`HERMES`), whose goal is to learn a representation such that individuals with "similar" treatment effects are brought closer together in latent space, while individuals with very different effects are distant by at least an *adaptive margin*.

We define a *contrastive loss* based on: *(i) positive pairs*, i.e., individuals with similar ITE value, and *(ii) negative pairs*, i.e., individuals with ITE values difference over a threshold $p$. In the rest of the paper, we will distinguish the *pair labels*, used by the Siamese network to say whether the pairs are similar with respect to the value of ITE, from the observed outcomes. In `HERMES`, pairs of units, i.e., the pair labels, are generated self-supervisedly from the model's current ITE estimates and are updated as training progresses: units with similar predicted treatment effects are treated as positive pairs, whereas units with dissimilar predicted effects are treated as negative pairs. Therefore, for a given pair of units, the label does not come from external annotation, but from the model's current estimate of their ITE similarity. To construct such pairs, no external similarity labels or ground-truth ITEs are assumed to be available. In parallel, the outcome heads are trained in a supervised way using the observed factual outcomes $Y$, from which these pseudo-ITE estimates are obtained. Unlike representation-learning approaches that organize the latent space mainly through global treated-control balance or propensity-based overlap, `HERMES` uses estimated treatment-effect similarity to shape the local geometry of the representation space, using a contrastive term which preserves overall moment-matching between treated and controls, while enforcing tight local alignment within positive pairs and margin-based separation for negative pairs.

On IHDP and Jobs benchmarks, `HERMES` outperforms `BCAUSS`, by leveraging local estimators that exploit the proximity of individuals in the latent space to produce more stable estimations.

The remainder of the paper is organized as follows. Section 2 introduces the problem setup and reviews the main representation-learning approaches for CATE estimation. Section 3 presents `HERMES`, including the Siamese architecture, the proposed contrastive objective, and its theoretical motivation. Section 4 describes the experimental setting, datasets, evaluation metrics, and implementation details. Section 5 reports the empirical results, including benchmark comparisons, latent-space analyses, and ablation studies. Finally, Section 6 concludes the paper and discusses the main implications of our findings.

## 2 Problem Setup and Related Work

We follow the potential outcomes framework of Rubin (1974) and consider a target population $P$ from which a realization of $n$ independent random variables $\{(X_i, T_i, Y_i(0), Y_i(1))\}_{i=1}^n \sim P$ is obtained as training data. Here, $X_i \in \mathbb{R}^d$ is a vector of $d$ pre-treatment covariates, $T_i \in \{0, 1\}$ is the treatment received (0 for control, 1 for treated), $Y_i(0) \in \mathbb{R}$ is the potential outcome for unit $i$ when assigned to the control group, and $Y_i(1) \in \mathbb{R}$ is the potential outcome for unit $i$ when assigned to the treatment group. We wish to make inference on the ITE, $D_i := Y_i(1) - Y_i(0)$, however, we only observe $Y_i = T_i Y_i(1) + (1 - T_i) Y_i(0)$, which highlights the main difference between ITE estimation and standard supervised learning.

Some inferential progress can be made by assuming *unconfoundedness*, also known as *strong ignorability of treatment assignment* (Rosembaum & Rubin, 1983),

$$\{Y(0), Y(1)\} \perp\!\!\!\perp T \mid X, \text{ and } 0 < Pr(T = 1 \mid X) < 1.$$

When the conditional means $E(Y(0) \mid X = x)$ and $E(Y(1) \mid X = x)$ exist, a more tractable estimand is the *conditional average treatment effect* (CATE),

$$\tau(x) = E\{Y(1) \mid X = x\} - E\{Y(0) \mid X = x\} = E\{Y(1) - Y(0) \mid X = x\},$$

which quantifies the expected treatment effect evaluated at a specific point in the feature space $x \in \mathcal{X}$. Crucially, the strong ignorability assumption is sufficient to identify the CATE (Künzel et al., 2019), and all that remains is to adapt statistical methods for the estimation of treatment effects to obtain CATE estimates with good asymptotic and finite sample properties.

## 2.1 Representation Learning for CATE Estimation

Access to a set of covariates that renders treatment assignment ignorable is essential for identifying causal effects from observational data. In practice, representation-learning methods can adjust only for information contained in the observed covariates, and therefore cannot rule out bias due to unmeasured confounders.

In most applications, however, one only has access to a vector of potential confounders $X$, and it is often more efficient to forego conditioning on the entire information in $X$, and instead work with a reduced dimensional representation $\Phi(X)$.

The goal is twofold: (i) the representation $\Phi(X)$ should be sufficient for predicting potential outcomes, and (ii) it should act as a deconfounding representation, i.e., conditioning on $\Phi(X)$ should render treatment assignment independent of the potential outcomes.

Formally, a representation function $\Phi : \mathcal{X} \to \mathcal{R}$, where $\mathcal{R}$ is the representation space, is a twice differentiable, one-to-one function. The function $\Phi$ pushes forward the treated and control distributions into $\mathcal{R}$; we denote the induced distribution by $p_\Phi$.

**Definition 1** *We indicate with $p_\Phi^{T=1}(r) = p_\Phi(r \mid T = 1)$ and $p_\Phi^{T=0}(r) = p_\Phi(r \mid T = 0)$ the treated and control distributions induced by the representation $\Phi$ over the latent space $\mathcal{R}$.*

Intuitively, $\Phi$ maps the original covariates $\mathcal{X}$ into $\mathcal{R}$, where each individual is represented by a vector $r = \Phi(x)$. The distributions $p_\Phi^{T=1}(r)$ and $p_\Phi^{T=0}(r)$ describe how the treated and control individuals are distributed in $\mathcal{R}$.

An important class of representation functions is defined in relation to the concept of balancing score (Rosembaum & Rubin, 1983).

**Definition 2 (Balancing score)** *A function $b(X)$ is called* balancing score *if : $X \perp\!\!\!\perp T \mid b(X)$.*

This implies that, among individuals with the same value of $b(X)$, the distribution of covariates $X$ is identical between the treated and control groups. Once the value of $b(X)$ is known, the treatment assignment $T$ provides no additional information about $X$. As a result, individuals with the same balancing score can be considered comparable, regardless of whether they received the treatment.

The best known case of balancing score is the *propensity score* $e(X) := P(T = 1 \mid X)$.

These types of representation functions play a crucial role in CATE estimation. Specifically, reminding ourselves that the observable outcome is denoted by $Y = T\,Y(1) + (1 - T)\,Y(0)$, we restate the classical result of (Rosembaum & Rubin, 1983).

**Theorem 1 (Rosenbaum & Rubin, 1983)** *If the information in $X$ ensures strong ignorability of treatment assignment and $b(X)$ is a balancing score, then for each fixed value $u$ of $b(X)$, we have:*

$$\underbrace{\mathbb{E}\{Y(1) \mid b(X){=}u\} - \mathbb{E}\{Y(0) \mid b(X){=}u\}}_{\text{causal effect at score } u} = \underbrace{\mathbb{E}\{Y \mid b(X){=}u,\, T{=}1\} - \mathbb{E}\{Y \mid b(X){=}u,\, T{=}0\}}_{\text{observed difference at score } u}.$$

The theorem states that within groups of individuals with the same balancing score, the difference in observed outcomes between treated and control individuals coincides with the causal effect of the treatment. Crucially,

while the left-hand side is expressed in terms of a counterfactual quantity, the right-hand side is expressed as the difference between observable quantities, and therefore amenable to estimation.

Beyond CATE, a common downstream estimand is the *average treatment effect* (ATE), defined by averaging over $X$, s.t.:

$$\psi = E[E\{Y \mid X, T=1\} - E\{Y \mid X, T=0\}] = E[E\{Y \mid b(X), T=1\} - E\{Y \mid b(X), T=0\}].$$

While balancing scores $b(X)$ summarize covariates through their relationship with treatment assignment $T$, ensuring $X \perp\!\!\!\perp T \mid b(X)$ (Rosembaum & Rubin, 1983), an alternative perspective is to construct reduced representations that retain the components of $X$ most relevant to potential outcomes and causal adjustment. A classical example is the *prognostic score* (Hansen, 2008), defined as a function $\Psi(X)$ such that, conditional on $\Psi(X)$, the covariates $X$ provide no additional information about a potential outcome. In the most common formulation, it is defined with respect to the untreated potential outcome, as follows:

$$Y(0) \perp\!\!\!\perp X \mid \Psi(X)$$

where usually $\Psi(X) = \mathbb{E}[Y(0) \mid X]$. Whereas balancing scores make units comparable with respect to treatment assignment, prognostic scores make them comparable with respect to baseline outcome risk.

Recent literature has moved beyond simple scalar propensity or prognostic summaries to explore low-dimensional, information-preserving representations. Sufficient dimension reduction techniques, for instance, aim to project original covariates into a compact space that strictly retains the information required for valid treatment-effect estimation. A prominent example is the *joint central subspace* (Huang & Chan, 2017), which is designed to capture the exact subset of $X$ governing both treatment assignment and the conditional distributions of potential outcomes. Building on this principle, related methodologies employ reduced covariate representations to facilitate robust matching and causal estimation, effectively circumventing the curse of dimensionality without compromising causal identification (Luo & Zhu, 2020).

This broader perspective has recently been formalized through the notion of *deconfounding scores* (D'Amour & Franks, 2021; Clivio et al., 2026). These are reduced representations $d(X)$ that preserve identification of the target causal estimand while potentially improving overlap relative to the original covariates.

From this viewpoint, propensity scores, prognostic scores, and other reduced representations can all be seen as members of a larger family of valid causal summaries. This perspective is particularly relevant because treatment-assignment information alone may contain instrumental components that do not improve outcome prediction and may even worsen finite-sample efficiency when conditioned upon (Hahn, 2004; Kuang et al., 2017). Targeted deconfounding representations aim to preserve the confounding-relevant and outcome-relevant structure of the covariates while discarding components that deteriorate overlap or inflate variance.

Our method follows this broader perspective, but differs in both the notion of similarity and the representation learned. Deconfounding-score representations preserve the information in $X$ needed for causal identification and confounding adjustment, yet they do not explicitly structure the latent space around treatment-response similarity. Similarly, prognostic representations group units by expected outcomes under a single treatment condition, which can support adjustment but does not necessarily imply similarity in treatment effects. Indeed, units with similar baseline risk may respond very differently to the same intervention, making prognostic similarity potentially inadequate for CATE estimation. `HERMES` addresses this limitation by learning representations organized around similarity in treatment response rather than expected outcomes under a single treatment condition. It introduces a Siamese representation-learning framework with self-supervised dynamic contrastive coupling and treatment-specific outcome heads for treated and control predictions. The contrastive signal is defined on their difference, $\hat{\tau}(x) = h_1(\Phi(x)) - h_0(\Phi(x))$, which acts as a model-based proxy for the ITE. Thus, similarity is explicitly grounded in an effect-relevant quantity rather than in single-treatment prognosis or unstructured deconfounding summaries.

The objective is therefore not merely to identify units with similar expected outcomes, but units that are likely to respond similarly to treatment. When counterfactual estimation relies on nearby representations, neighborhoods organized around treatment-response similarity can be more informative than those based solely on prognosis. For instance, two high-risk patients may be close in a prognostic space, even if only

one substantially benefits from treatment. While a prognostic approach would tend to group them together, HERMES aims to structure the latent space so that local neighborhoods reflect similarity in predicted treatment response, making the representation more directly informative for fine-grained CATE estimation.

Methods such as *back-door adjustment* (Belloni et al., 2013; Chernozhukov et al., 2024) and *weighting* approaches (Austin, 2011) use the propensity score to re-weight observational data, ensuring comparability between treated and control groups. *Targeted Learning* (van der Laan & Rose, 2018) refines initial estimators through a second-stage model to achieve an optimal bias–variance trade-off, while other approaches design treatment-effect–specific splitting criteria (Su et al., 2009; Zhang et al., 2017) or leverage ensemble/meta-algorithms (Künzel et al., 2019; Wager & Athey, 2017). Neural network–based solutions, such as *Propensity Dropout* (Alaa et al., 2017) and *Perfect Matching* Schwab et al. (2019), integrate propensity scores into their architectures. Here, we focus on *representation learning methods* through the formalism introduced by Shalit et al. (2017).

## 2.2 Representation Functions and Errors in Estimation

Let $\Phi(X)$ be a representation function and denote the conditional expected outcome with $h_t\{\Phi(x)\} := E\{Y \mid T = t, \Phi(X) = \Phi(x)\}$. If $\Phi(X)$ is a balancing score, and $\hat{h}_t : \mathcal{T} \times \mathcal{X} \to \mathcal{Y}$ is an estimator of the conditional outcome, we may consider CATE estimators of the form:

$$\hat{\tau}_\Phi(x) = \hat{h}_1\{\Phi(x)\} - \hat{h}_0\{\Phi(x)\}.$$

Crucially, even under balancing requirements, the representation function $\Phi(X)$ is not unique. We are, therefore, interested in learning $\Phi(X)$ to minimize the expected mean square error (EMSE):

$$\text{EMSE}_\Phi = E\left[\{\tau(x) - \hat{\tau}_\Phi(x)\}^2\right].$$

In this context, a fundamental framework for representation learning in causal inference was introduced by Shalit et al. (2017), who developed an upper bound for the EMSE, which can be expressed as the sum of factual losses and a measure of distributional imbalance between treated and control subpopulations.

Let $L : \mathcal{Y} \times \mathcal{Y} \to \mathbb{R}_+$ be a *loss function*. The expected factual treated and control losses are defined as:

$$\varepsilon_F\left\{\hat{h}_1(\Phi)\right\} := E\left[L\left\{Y, \hat{h}_1(\Phi(X))\right\} \mid T = 1\right], \qquad \varepsilon_F\left\{\hat{h}_0(\Phi)\right\} = E\left[L\left\{Y, \hat{h}_0(\Phi(X))\right\} \mid T = 0\right].$$

The term $\varepsilon_F\{\hat{h}_1(\Phi)\}$ represents the expected predictive error restricted to the treated group. Analogously, $\varepsilon_F\{\hat{h}_0(\Phi)\}$ captures the expected predictive error restricted to the control group. Together, these two losses provide a group-specific decomposition of the overall factual risk.

Distributional imbalance may be quantified using *Integral Probability Metrics* (IPM). Specifically, for two probability density functions $p, q$ defined over $S \subseteq \mathbb{R}^d$, and for a family $\mathcal{G}$ of functions $g : S \to \mathbb{R}$, we define the IPM between probability distributions as:

$$\text{IPM}_\mathcal{G}(p, q) = \sup_{g \in \mathcal{G}} \left| \int_S g(s)\big(p(s) - q(s)\big)\, ds \right|.$$

The choice of the function class $\mathcal{G}$ determines the specific form of the metric. For example, if $\mathcal{G}$ is the set of 1-Lipschitz functions, the IPM reduces to the Wasserstein distance; if $\mathcal{G}$ is the unit ball of a reproducing kernel Hilbert space (RKHS), the IPM reduces to the well-known *Maximum Mean Discrepancy* (MMD).

Below we restate a simplified version of the result by Shalit et al. (2017), which will serve as a guiding principle for the design of CATE estimation strategies.

**Theorem 2** *Let $\Phi : \mathcal{X} \to \mathcal{R}$ be a one-to-one representation function, $\sigma_Y^2$ be a residual variance parameter, and $\mathcal{G}$ be a suitable family of functions. Then, under squared error loss, there exists a constant $B_\Phi > 0$ s.t.:*

$$EMSE_\Phi \leq 2\left[\varepsilon_F\left\{\hat{h}_1(\Phi)\right\} + \varepsilon_F\left\{\hat{h}_0(\Phi)\right\} + B_\Phi \cdot \text{IPM}_\mathcal{G}\big(p_\Phi^{T=1}, p_\Phi^{T=0}\big) - 2\sigma_Y^2\right], \tag{1}$$

Theorem 2 establishes that minimizing the EMSE requires two complementary objectives: achieving low prediction error on the observed outcomes for both treated and control individuals, and ensuring that the learned representation aligns the distributions of treated and untreated populations in the representation space, as measured by the IPM.

## 2.3 CATE and Deep Neural Representation Learning

The full expressivity of deep learning strategies has proven to be of crucial importance in the estimation of the CATE function $\tau(x)$. One of the earliest contributions in this domain is the `TARNet` structure introduced by Shalit et al. (2017) and its refined counterpart (`CFRNet`) (Shalit et al., 2017).

Both methods obtain stable estimates of the conditional outcome functions $h_1(\Phi)$ and $h_0(\Phi)$ by sharing a common representation of the covariates. A "Y-shaped" architecture is proposed; consisting of that feeds two separate potential-outcome heads, $h_0$ and $h_1$. The encoder maps the input covariates $X$ to a latent representation $\Phi(X)$. The two heads then take this representation to estimate the potential outcomes: $\hat{Y}(1) = h_1(\Phi(X))$ and $\hat{Y}(0) = h_0(\Phi(X))$.

`TARNet` is trained end-to-end to minimize the regression error for *factual* (observed) outcomes. While deploying the same architecture, `CFRNet` encourages balanced representations by introducing a loss function, which adds a regularization term to the factual loss and explicitly penalizes the discrepancy between distributions, measured as:

$$\mathcal{L}_{\text{CFR}} = \mathcal{L}_{\text{factual}} + \alpha \cdot \text{MMD}^2\{\Phi(\mathbf{X}_{T=1}), \Phi(\mathbf{X}_{T=0})\}, \tag{2}$$

where $\mathcal{L}_{\text{factual}}$ is the loss function of `TARNet`, MMD is the *Maximum Mean Discrepancy*, and $\alpha$ is a hyperparameter controlling the strength of the balancing constraint. By forcing the latent distributions to overlap, `CFRNet` mitigates the out-of-distribution problem that plagues `TARNet`. However, this global alignment may be ineffective, since MMD operates at an aggregate level, ensuring the overall distributions match, but it provides no guarantee of *local* balance.

While `CFRNet` addresses observed covariate imbalance by encouraging balanced latent representations, other deep learning architectures have aimed more explicitly at learning representation functions with balancing-score-like properties. `DragonNet` (Shi et al., 2019), for example, extends the "Y-shaped" architecture into a "trident" by adding a third head dedicated to estimating a propensity score. In this framework, training is based on loss functions aimed at ensuring asymptotic efficiency and double robustness through targeted regularization (van der Laan & Rose, 2018). A similar approach was adopted in Tesei et al. (2023) - `BCAUSS`, which uses the architecture of `DragonNet` but learns a *balancing score* $g(\mathbf{x})$ through a loss function $\mathcal{L}_{\text{BAL}}$ which minimizes the squared difference between the weighted means of the covariates in treated $(t_i)$ and control $(1 - t_i)$ groups:

$$\mathcal{L}_{\text{BAL}} = \frac{1}{d} \sum_{k=1}^{d} \left( \frac{\sum_{i=1}^{n} \frac{t_i}{g(\boldsymbol{x}_i)} x_{i,k}}{\sum_{i=1}^{n} \frac{t_i}{g(\boldsymbol{x}_i)}} - \frac{\sum_{i=1}^{n} \frac{1-t_i}{1-g(\boldsymbol{x}_i)} x_{i,k}}{\sum_{i=1}^{n} \frac{1-t_i}{1-g(\boldsymbol{x}_i)}} \right)^2. \tag{3}$$

Here, direct optimization in the balance of pretreatment covariates proves to enhance robustness in empirical scenarios with poor distributional overlap (Fan et al., 2023).

The idea of overlapping (positive) vs. non-overlapping (negative) samples has also been exploited using contrastive learning approaches, e.g. in the *Contrastive Individual Treatment Effects* (`CITE`) method proposed by Li & Yao (2022). Its core idea is to use the propensity score to define *positive* (individuals with propensity scores near 0.5, i.e., the "ideal" balanced case) and *negative* (individuals with scores near 0 or 1. i.e., the sources of bias) examples. Using such pairs, a noise contrastive loss forces the model to push the representations of positive pairs closer together (attraction) while simultaneously pushing apart the representations of negative pairs (repulsion) in the latent space. While potentially appealing, this approach is entirely dependent on an external, auxiliary, pre-trained propensity score model, which may influence the quality of the final learned representation.

While both `CITE` and `HERMES` rely on contrastive learning, they organize the latent space according to different principles. In `CITE`, contrastive signal is designed to favor regions of better overlap, that is, regions

where treatment assignment is less deterministic from the representation. This can be beneficial for reducing imbalance between treated and control groups, but it does not necessarily imply that nearby units will also have similar treatment effects, since good overlap alone does not guarantee local homogeneity in the ITE. By contrast, `HERMES` defines positive and negative pairs according to similarity in the estimated ITE. The contrastive objective therefore encourages units with similar predicted treatment effects to be close in the latent space, making local neighborhoods more informative for individual causal-effect estimation. At the same time, `HERMES` does not ignore selection bias: by bringing together treated and control individuals with similar treatment effects, it also promotes representations in which treatment assignment becomes less informative within local neighborhoods. In this sense, `HERMES` is designed not only to improve overlap, but also to organize the latent space around expected causal response. This distinction becomes particularly relevant in low-overlap regions. In `CITE`, units in such regions are mainly used as negative contrastive examples, so the method encourages the representation to move away from highly imbalanced areas, even though treatment-effect predictions must still be produced there. `HERMES`, instead, aims to make those local comparisons more meaningful for CATE estimation by structuring the representation around treatment-effect similarity rather than around overlap alone. Beyond functioning as a standalone architecture, `HERMES` serves as a modular contrastive add-on for existing causal models. For any CATE estimator providing treatment-specific predictions, our dynamic ITE-based pairing strategy can be applied to regularize its latent space. Our core contribution is therefore methodological: we introduce a novel causal role for contrastive learning, organizing representations based on treatment-response similarity rather than simply enforcing overlap.

Within this contextual setting, our work aims to address the central challenge posed by CATE estimation: *unreliable extrapolation in regions of poor local support.* This occurs when a model must predict a treatment effect for an individual without having similar "counterfactual twins" in the comparison group, forcing it to make risky assumptions based on general trends. In the following sections, we introduce `HERMES`, a framework that combines self supervised and contrastive learning, for globally and locally balanced representations of covariate information, with the goal of improving local comparability for CATE estimation.

## 3 CATE Representation Learning via Matched Siamese Embeddings

Theorem 2 shows that reducing the EMSE requires *both (i)* low factual prediction errors and *(ii)* a small discrepancy between the latent distributions of treated and controls. Our hypothesis is that we can further reduce representational discrepancies by introducing a *contrastive loss* function. The intuition is to force the model to create a latent space $\mathcal{R}$ in which the notion of "distance in representations" is semantically linked to "difference in treatment effect". The IPM measures the discrepancy between the distributions in a latent space. As explained above, an important and computationally tractable class of IPMs is the MMD, which measures the distance between the means of the two distributions mapped in a high-dimensional space (MMD of value zero if the distributions are identical).Our idea is to define a contrastive loss function that encourages a structured form of local alignment in the latent space, beyond purely global MMD-style matching. We integrated this contrastive loss within a Siamese Neural Network, namely `HERMES`, consisting of two twin branches, where each branch is a `BCAUSS` architecture. The two branches share all parameters and process input pairs while applying the contrastive loss in a shared latent space structured around treatment-effect similarity. The contrastive loss operates on *positive* (resp. *negative*) pairs, i.e., individuals with similar (resp. dissimilar) ITEs. It encourages the model to bring positive pairs closer together in the latent space $\mathcal{R}$ while pushing negative pairs apart beyond a specified margin. This induces a semantic structure in $\mathcal{R}$ where distances reflect differences in treatment effect rather than mere similarity in covariates and prevents the representation space from being shaped solely on the latter (see Figure 1).

The proposed contrastive loss has dynamic behavior: pairs' labels (similar or not similar) are recomputed and refined at each training epoch according to the model's current ITE estimates. This self-supervised mechanism allows the latent representation to co-evolve as the model's predictive ability improves. This leads to local neighborhoods that are progressively refined and populated by individuals who, despite having similar ITE values, may come from both treatment and control groups. This can support more reliable local counterfactual comparisons for CATE estimation as it favors the presence of a close "counterfactual twin" for each individual. The learned latent space aims to create regions of improved local overlap where individuals with opposite treatment are close together, leading to a decrease in MMD, and shifting the extrapolation

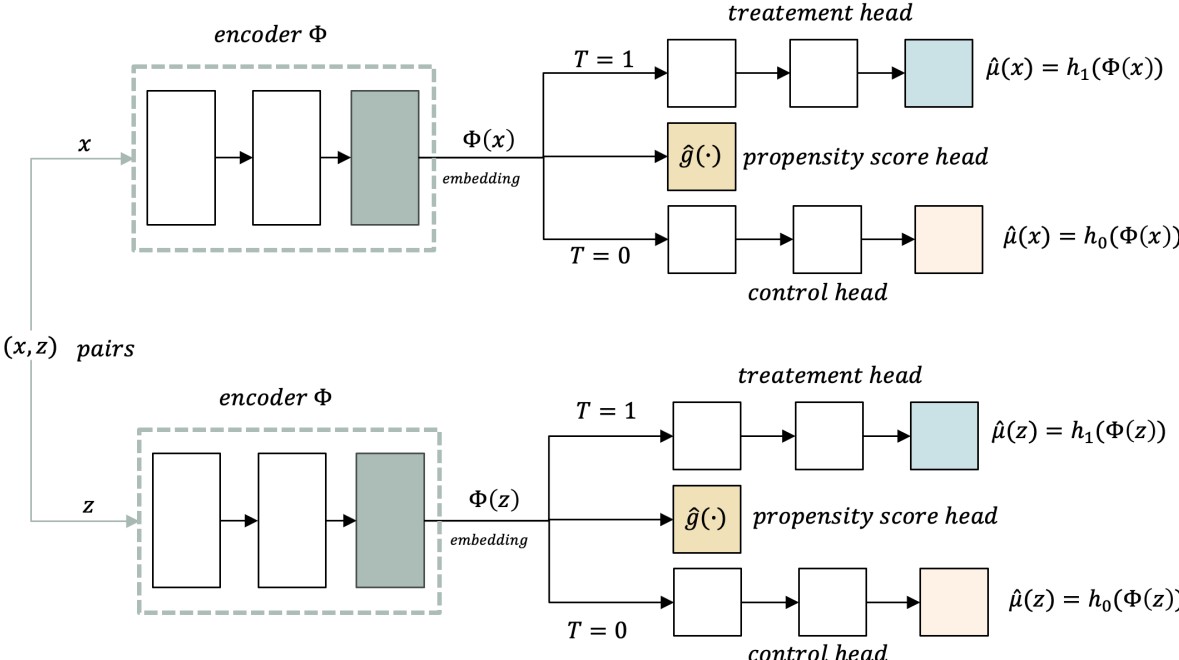

Figure 1: Overview of HERMES: a pair $(x, z)$ of covariate vectors is passed through two branches; each branch consists of an *encoder* $\Phi(\cdot)$ that yields the latent embedding $\Phi(x)$, two *treatment-specific heads* to estimate the potential outcomes $\hat{\mu}_T(x) = h_T(\Phi(x))$, and a third head which calculates the propensity score $\hat{g}(\cdot)$; the output of the network is an ITE-similarity score between $x$ and $z$.

problem towards a more stable interpolation problem. We remark that HERMES does not assume that effect-homogeneous clusters are available a priori. Rather, the method uses the model's current ITE predictions only as an internal self-supervised signal to identify pairs that are more likely to have similar treatment effects than random alternatives. The neighborhood structure is therefore not fixed before training, but progressively induced and refined as the representation improves.

## 3.1 Network Architecture and Loss Functions

HERMES takes as input a pair of covariate vectors $(x, z)$. Each vector is in turn passed to one of the two twin branches, each one adopting the following architecture: an encoder $\Phi(\cdot)$, built as a three-layer MLP $\Phi : \mathcal{X} \to \mathcal{R}$ (ReLU, 200 neurons per layer) maps the input vector $v$ ($x$ or $z$) to an embedding $\Phi(v)$; from this embedding, two separate treatment-specific "heads" $h_0(\Phi(v))$ and $h_1(\Phi(v))$, with $h_0, h_1 : \mathcal{R} \to \mathbb{R}$, estimate the potential outcomes $\hat{\mu}_T(v) = h_T(\Phi(v))$, $T \in \{0, 1\}$, for control ($T = 0$) and treated ($T = 1$) individuals, respectively; each head is a two-layer MLP with ReLU and hidden dimension width $N/2$ where $N$ is number of neurons of the previous layer, followed by a linear layer; a third head $g : \mathcal{R} \to (0, 1)$, i.e., a single linear unit with sigmoid activation function on the shared representation, produces the propensity score $g(\Phi(v)) = \sigma(w^\top \Phi(v) + b)$.

The treatment-specific heads $h_0$ and $h_1$ can be interpreted as learned prognostic functions under control and treatment, respectively. Crucially, however, the self-supervised pairing rule in HERMES is not defined on either prognostic function in isolation, but on their contrast $\hat{\tau}(x) = h_1(\Phi(x)) - h_0(\Phi(x))$ which provides a model-based proxy for treatment-effect similarity. This yields a different organizing principle from that of classical prognostic scores, which summarize expected outcome risk under a single treatment condition, most commonly through $Y(0)$ or $\mathbb{E}[Y(0) \mid X]$, and are therefore primarily prognosis-oriented. By contrast, HERMES uses the discrepancy between treatment-specific outcome predictions to structure local neighborhoods according to estimated treatment response. This proxy is not treated as a fixed estimate of the true ITE, but as a progressively refined signal that guides the self-supervised pairing mechanism during training.

Each branch is trained with a composite loss function $\mathcal{L}_{\text{branch}}$ defined as follows:

$$\mathcal{L}_{\text{branch}} = \mathcal{L}_{\text{base}} + \lambda_{\text{bce}}\,\mathcal{L}_{\text{bce}} + \lambda_{\text{bal}}\,\mathcal{L}_{\text{bal}} + \lambda_{\text{targ}}\,\mathcal{L}_{\text{targ}} + \lambda_{\ell_2}\,\|\theta\|_2^2 \tag{4}$$

- $\mathcal{L}_{\text{base}}$ is the mean squared error between the observed outcome $y$ and its factual prediction $y_{\text{pred}} = T \cdot \hat{h}_1(\Phi(v)) + (1 - T) \cdot \hat{h}_0(\Phi(v))$; this term contributes to reducing the factual component of the *EMSE* bound defined in Theorem 2;

- $\mathcal{L}_{\text{bce}}$ is a binary cross-entropy loss on the treatment indicator, computed from a "smoothed" propensity score $\tilde{g}(\Phi(v)) = \frac{\hat{g}(\Phi(v)) + 10^{-3}}{1.002}$, which never reaches values arbitrarily close to 0 or 1, thus avoiding degenerate inverse-propensity weights and promoting stable learning; formally, for a treatment assignment $T \in \{0, 1\}$ and predicted probability $\tilde{g}(\Phi(v))$, the binary cross-entropy loss $\mathcal{L}_{\text{bce}}(v, T)$ (or $\mathcal{L}_{\text{bce}}$) is defined as:

  $$\mathcal{L}_{\text{bce}}(v, T) \;=\; -\Big(T \log \tilde{g}(\Phi(v)) + (1 - T) \log\big(1 - \tilde{g}(\Phi(v))\big)\Big)$$

  this term encourages correct separation of treated and control individuals.

- $\mathcal{L}_{\text{bal}}$ is a feature-balancing penalty (see Equation 3) that enforces the overlap between treated and control groups by matching inverse-propensity–weighted covariate means; specifically, each individual is reweighted with $w_1 = \frac{T}{\tilde{g}(\Phi(v))}$ and $w_0 = \frac{1 - T}{1 - \tilde{g}(\Phi(v))}$, so that the covariate distribution of treated and controls is aligned;

- $\mathcal{L}_{\text{targ}}$ is a regularizer that enforces coherence between outcome and propensity score predictions (and so the treatment assignment process); the idea is that, in regions of "low overlap", i.e., where treated individuals have very low propensity or controls have very high propensity, the treatment and control head models are learning to associate the outcome with the covariates rather than with the treatment effect; we would force the model to align its outcome predictions with both the observed outcome and the estimated propensity score; to this aim, we introduce a perturbed prediction $y_{\text{pert}} = y_{\text{pred}} + \varepsilon\,\gamma(x, T)$ where $\gamma(x, T) = \frac{T}{\tilde{g}(\Phi(v))} - \frac{1 - T}{1 - \tilde{g}(\Phi(v))}$ is the *assignment direction*, $\varepsilon \sim \text{Unif}[0, 1)$, and define $\mathcal{L}_{\text{targ}} = (y - y_{\text{pert}})^2$; $\gamma(x, T)$ quantifies how "unexpected" the treatment assignment is given the covariates; for treated individuals, we have $\gamma(x, 1) = 1/\tilde{g}(\Phi(x))$, and this value becomes large when the estimated propensity score $\tilde{g}(\Phi(x))$ is small, i.e., when the individual was unlikely to be treated; for control individuals, we have $\gamma(x, 0) = -1/(1 - \tilde{g}(\Phi(x)))$, and this value becomes large in magnitude when the individual was unlikely to remain untreated; hence, $|\gamma(x, T)|$ is large in regions of low overlap; by integrating $(y - y_{\text{pert}})^2$ in $\mathcal{L}_{\text{branch}}$, we shift the factual prediction along this assignment direction, i.e., we are penalizing the outcome prediction in low overlap regions to align it with the propensity score prediction.

- *(v)* the $\ell_2$ regularization term $\lambda_{\ell_2}\|\theta\|_2^2 \;=\; \lambda_{\ell_2} \sum_{W \in \mathcal{W}}\|W\|_F^2$ applies weight decay to the network parameters; here, $\theta$ denotes the full set of learnable parameters of the model, $\mathcal{W} \subset \theta$ is the subset consisting of weight matrices, and $\|W\|_F^2 = \sum_{i,j} W_{ij}^2$ is the squared Frobenius norm of a weight matrix $W$; the effect of this term is equivalent to augmenting the loss value with a quadratic penalty on the weights, i.e., $\lambda_{\ell_2}\|\theta\|_2^2$; intuitively, this discourages excessively large weights, reduces overfitting by controlling the effective capacity of the network; in the context of HERMES, weight decay plays an additional role: it keeps the scale of the latent representations $\Phi(x)$ under control, ensuring that distances in the embedding space remain consistent with the contrastive margin $m$; this prevents degenerate solutions in which embeddings grow unboundedly just to satisfy the margin constraint, and guarantees that the latent space geometry remains meaningful for treatment-effect estimation.

Now, we provide details about the contrastive loss used to train HERMES, which is built using the loss functions of the two twin branches defined above. Intuitively, let $\texttt{branch}_x$ and $\texttt{branch}_z$ be the two twin branches of the HERMES model, the final loss function is obtained by adding $\mathcal{L}_{\text{branch}_x}$ with $\mathcal{L}_{\text{branch}_z}$, and with a contrastive loss term explained in detail below.

The training process relies on self-supervised pairwise labeling based on the model's current ITE predictions. First, we provide some basic definitions that will help explain the training steps below. We remark that, each individual actually consists of a triplet $(x, T, y)$ where $x$ is the vector of covariates, $T$ indicates whether the individual was treated $(T = 1)$ or not $(T = 0)$, and $y$ is the corresponding outcome. Let's consider a dataset $\mathcal{D}$ of covariate vectors used to construct the pairs that will be given as input to the model during the training process, we say that any $\mathcal{M} \subset \mathcal{D} \times \mathcal{D}$ is a *mini-batch* of $\mathcal{D}$. For each covariate vector pair $(x, z) \in \mathcal{D} \times \mathcal{D}$, we define $\Delta\tau_{xz} = |\hat{\tau}(x) - \hat{\tau}(z)|$, where $\hat{\tau}(x) = \hat{\mu}_1(\Phi(x)) - \hat{\mu}_0(\Phi(x))$ and $\hat{\tau}(z) = \hat{\mu}_1(\Phi(z)) - \hat{\mu}_0(\Phi(z))$. The training consists in repeating the following phases for $E$ epochs:

1. Build a mini-batch $\mathcal{M} = \{(x_i, z_i)\}_{i=1}^m \subset \mathcal{D} \times \mathcal{D}$, with $x_i \neq z_i$ for $i = 1, \ldots, m$, by randomly selecting $m$ pairs from $\mathcal{D} \times \mathcal{D}$.

2. Compute $\Delta\tau_{x_i z_i}$ for all $(x_i, z_i) \in \mathcal{M}$ by using the current predicted ITEs.

3. Let $\tau_{\mathrm{thr}}$ be the similarity threshold, set the *positive pairs* $\mathcal{P}_{pos} = \{(x, z) \in \mathcal{M} : \Delta\tau_{xy} \leq \tau_{\mathrm{thr}}\}$ and the set *negative pairs* $\mathcal{P}_{neg} = \mathcal{M} \setminus \mathcal{P}_{pos}$; these labels are frozen until next labeling phase.

4. Let build a *labeled* mini-batch $\mathcal{M}_l$ by randomly selecting $k$ pairs from $\mathcal{P}_{pos}$ (pairs labeled with 1) and $k$ pairs from $\mathcal{P}_{neg}$ (pairs labeled with 0).

5. Let compute $\mathcal{L}_{\mathrm{tot}} = \mathcal{L}_{\mathrm{branch_x}} + \mathcal{L}_{\mathrm{branch_z}} + \lambda\,\mathcal{L}_{ctr}$, where $\lambda > 0$, $\mathcal{L}_{ctr} = \sum_{(x,z)\in\mathcal{M}_l} \mathcal{L}_{ctr}(\Phi(x), \Phi(z))$, with $\mathcal{L}_{ctr}$ which encourages positive pairs to be closer than a predefined margin $m$ in the latent space and penalizes the remaining negative pairs:

$$\mathcal{L}_{ctr}(\Phi(x), \Phi(z)) = \begin{cases} \|\Phi(x) - \Phi(z)\|_2^2, & \text{if } \Delta\tau_{xz} \leq \tau_{\mathrm{thr}}, \\ \max\{0,\, m - \|\Phi(x) - \Phi(z)\|_2\} & \text{otherwise.} \end{cases}$$

We remark that we adopt a short *pre-training* phase in which the term $\mathcal{L}_{ctr}$ is not included in $\mathcal{L}_{tot}$. This is because, before activating the contrastive loss, we improve the initial ITE prediction ability for the first pairs. Finally, the loss function $\mathcal{L}_{\mathrm{tot}}$ is used to train the model. Observe that minimizing $\mathcal{L}_{\mathrm{tot}}$ forces the model to satisfy *both* goals of Theorem 2: low factual error and small treated-vs-control discrepancy. See Algorithm 1 for details about the `HERMES` training process formalized above.

Repeat steps 4-5 for $E_l$ steps, then return to step 1, so as not to have to calculate the ITE in each epoch, reducing, on the one hand, the computational effort, and on the other hand, allowing the model to stabilize the prediction ability of the ITE.

## 3.2 Contrastive Regularization and IPM Reduction

The generalization error bound of Shalit et al. (2017) relies on the fact that the counterfactual loss can be upper bounded by the sum of the factual prediction error and a distributional discrepancy term expressed as an IPM between the representations of treated and control groups. Intuitively, $\mathrm{IPM}\big(p_\Phi^{T=1}, p_\Phi^{T=0}\big)$ measures how far apart the two groups remain in the latent space $\mathcal{R}$. Minimizing this global discrepancy is the central objective of methods such as `CFRNet` and `BCAUSS`. However, this strategy only enforces *average alignment* between distributions, which may still leave significant local mismatches: two individuals with highly similar treatment effects might remain distant in the latent space if their raw covariates differ substantially. This phenomenon forces the model to extrapolate across regions where local overlap is poor, inflating estimation error.

The contrastive regularization in `HERMES` reconstructs the latent space around ITE similarity. Rather than relying solely on covariate-based alignment, the contrastive loss encourages embeddings of individuals with similar estimated ITEs to cluster together, regardless of their treatment assignment. Negative pairs are pushed apart by a margin $m$. This encourages the representation space to organize into locally effect-homogeneous regions, in which units with similar estimated treatment effects are mapped nearby, while units with dissimilar effects are kept separated. Within each cluster, treated and control individuals become nearly indistinguishable in distribution, while between clusters the separation is enforced.

This intuition can be formalized by decomposing the global discrepancy into *intra-cluster* and *inter-cluster* contributions. Let $p_\Phi^{T=1,g}$ and $p_\Phi^{T=0,g}$ be the conditional distributions of treated and controls restricted to cluster $g$, with mixture weights $\pi_g$, the decomposition is defined as follows:

$$\text{IPM}\big(p_\Phi^{T=1}, p_\Phi^{T=0}\big) \ \leq \ \sum_g \pi_g \ \text{IPM}\big(p_\Phi^{T=1,g}, p_\Phi^{T=0,g}\big) \ + \ \delta(m) \tag{5}$$

where the first term measures the residual discrepancy *within* clusters and $\delta(m)$ quantifies the separation induced between clusters, controlled by the contrastive margin $m$.

The coefficients $\pi_g$ denote the *mixture weights* associated with each cluster $C_g$. Formally, $\pi_g$ is the probability that a representation falling into cluster $C_g$. Thus, the global treated distribution can be written as $p_\Phi^{T=1}(r) = \sum_g \pi_g \, p_\Phi^{T=1,g}(r)$ and analogously for the control distribution. The weights $\pi_g$ reflect the relative importance of each cluster when reconstructing the overall treated and control distributions from their local counterparts.

The key insight is that in `HERMES` the intra-cluster IPM terms are explicitly minimized by the contrastive mechanism, since positive pairs enforce similarity of treated and controls with comparable ITEs. The remaining discrepancy across clusters does not harm estimation: by design, individuals in different clusters are expected to have substantially different treatment effects, so attempting to balance them globally would instead introduce extrapolation bias. The margin $m$ ensures that such clusters remain well separated, effectively confining the estimation problem to regions where overlap is meaningful. We remark that the difference between classical approaches such as `BCAUSS` and the proposed `HERMES`. In `BCAUSS`, the only controllable quantity is the *global* IPM between treated and control distributions. This metric may remain large, since it is sensitive to differences averaged across the entire representation space. In contrast, `HERMES` enforces an internal structure by splitting the representation space into positive and negative clusters according to ITE similarity. Within each cluster, treated and control distributions are aligned and become nearly indistinguishable, yielding a much lower *local IPM*, which is the quantity that truly matters for interpolation-based CATE estimation. Differences between clusters, on the other hand, are not problematic: when ITE values are very different, aligning treated and controls would only induce extrapolation. The contrastive margin $m$ ensures that such regions remain well separated. Consequently, while the global IPM may remain non-negligible, the *effective IPM* relevant for causal estimation is significantly reduced.

**Theorem 3 (Contrastive IPM Bound)** *Let $\Phi : \mathcal{X} \to \mathcal{R}$ be a one-to-one representation function, with inverse $\Psi$, and $h$ the conditional expectation. Suppose the representation space $\mathcal{R}$ admits a partition into clusters $C_g$ corresponding to locally effect-homogeneous regions induced by the contrastive objective. Let $p_\Phi^{T=t,g}$ denote the distribution of representations for group $T = t$ restricted to cluster $C_g$, with mixture weights $\pi_g$. Then, under the same assumptions as Theorem 2, the counterfactual loss admits the bound*

$$EMSE_\Phi \ \leq \ 2\left[\varepsilon_F\left\{\hat{h}_1(\Phi)\right\} + \varepsilon_F\left\{\hat{h}_0(\Phi)\right\} + B_\Phi \cdot \left(\sum_g \pi_g \ \text{IPM}_\mathcal{G}\big(p_\Phi^{T=1,g}, p_\Phi^{T=0,g}\big) + \delta(m)\right)\right],$$

*where $\delta(m)$ decreases monotonically with $m$ and vanishes when clusters are perfectly separated.*

This result shows that the contrastive loss does not simply reduce the global IPM in aggregate, but rather enforces a more structured decomposition where treated and control individuals are aligned *within* clusters of similar ITE, while the margin $m$ ensures that different clusters remain well separated. The overall effect is to replace the original global discrepancy with a more informative local discrepancy that is easier to minimize and better aligned with the causal estimation task. This refinement provides intuition for why `HERMES` may achieve smaller effective discrepancy in practice, effectively transforming the problem of global alignment into one of local, semantically meaningful interpolation.

Theorem 3 is not intended as a uniformly tighter version of Theorem 2. Instead, it refines the global discrepancy term by decomposing it into cluster-wise IPM contributions together with a residual term $\delta(m)$ induced by the margin. The advantage of this refinement appears in the regime where the contrastive objective successfully organizes the representation space into clusters that are locally homogeneous with respect to treatment effects. In that case, the within-cluster discrepancies $\text{IPM}_\mathcal{G}\big(p_\Phi^{T=1,g}, p_\Phi^{T=0,g}\big)$ can be

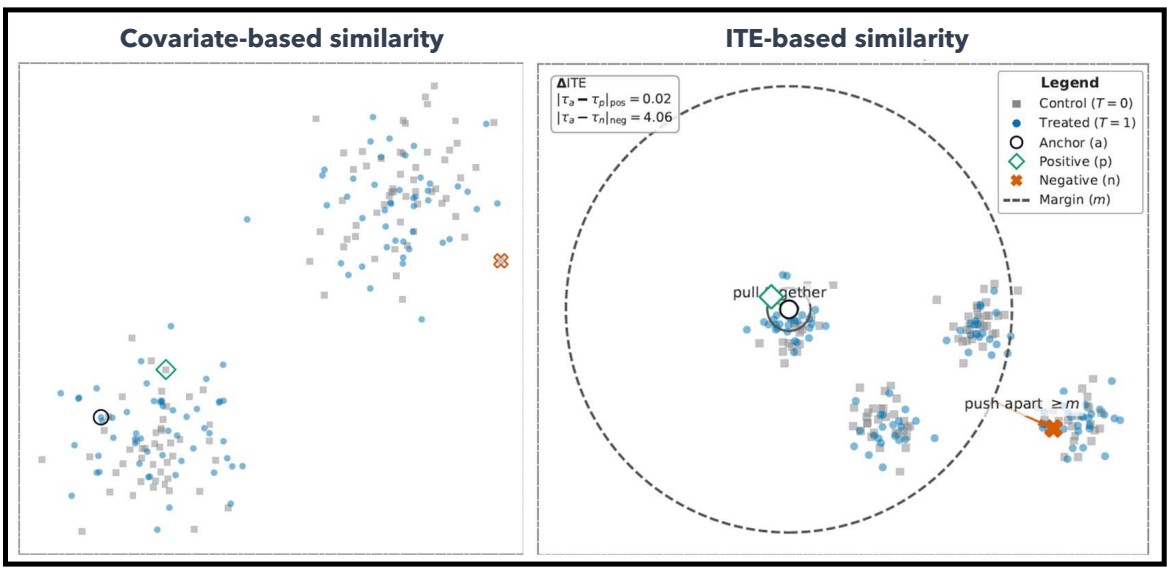

Figure 2: *Effect of HERMES's Contrastive Loss on the Latent Space structure:* when the representation is organized by covariate similarity, individuals with comparable ITE may be distributed far apart, forcing the model to rely on extrapolation; in contrast, HERMES organizes the latent space by ITE similarity, i.e., positive pairs with similar effects are pulled close together, while negative pairs are pushed apart by at least the margin $m$; this creates locally dense and causally meaningful neighborhoods, turning extrapolation into a more reliable interpolation problem.

substantially smaller than the corresponding global IPM, since alignment is required only within locally similar regions rather than across the entire population. At the same time, the residual cross-cluster mismatch is controlled by the margin parameter through $\delta(m)$. Therefore, Theorem 3 becomes advantageous precisely when local alignment is easier than global alignment, which is the setting targeted by HERMES. In particular, whenever the sum of the cluster-wise discrepancies plus $\delta(m)$ is smaller than the global IPM term appearing in Theorem 2, Theorem 3 provides a more informative characterization of the estimation error.

The cluster decomposition in Theorem 3 should be interpreted as an idealized description of the representation geometry induced by training, rather than as an observed partition available *a priori*. In HERMES, such local groupings are not assumed to exist from the start. Rather, they are expected to emerge progressively as the factual objective first extracts a weak effect-relevant signal and the contrastive term subsequently reshapes the latent space around it. For this reason, Theorem 3 should not be read as a full convergence or stability guarantee for the noisy self-supervised pairing dynamics. Since pair assignments are constructed from the model's current pseudo-ITE estimates, they are inevitably subject to estimation error, especially in the early stages of training. The role of the theorem is instead to clarify why local effect-aware alignment can be preferable to global balancing once a meaningful effect-relevant structure begins to emerge. The practical stability of this feedback process is addressed separately through the training design of HERMES—namely warm-up, periodic relabeling, and persistent factual supervision—and is evaluated empirically in the stress tests reported in Appendix B.3.

A graphical comparison of the induced latent spaces is shown in Figure 2. On the left, when the representations are distributed according to *covariate similarity*, individuals with comparable treatment effects but heterogeneous covariate profiles may be embedded far apart. For example, the anchor individual $a$ (empty circle with black border) and its counterfactual counterpart $p$ (empty rhombus with green border) share a similar ITE but remain distant in the latent space because their covariates have different distribution. This spatial separation forces the model to extrapolate across regions unsupported by reliable counterfactual evidence. On the right, the proposed HERMES model organizes the representation space by *ITE similarity*. Here, the contrastive loss actively pulls together positive pairs such as $a$ and $p$, ensuring that treated and control

---

**Algorithm 1:** Training procedure of `HERMES` with contrastive labeling.

---

**Input** : Dataset $\{(x_i, T_i, y_i)\}_{i=1}^n$, hyperparameters $m, \lambda, E, B, k$
**Output:** Trained HERMES model

**1** Initialize model weights $h(\Phi(\cdot), T)$;
**2** (Optional) Pre-train with $\mathcal{L}_{\text{branch}}$ only for a few epochs;
**3** **for** epoch$Gets$1 **to** $E$ **do**
    // Pairing Update Phase (done once per epoch)
**4**    Compute ITE estimates $\hat{\tau}(x) = \mu_1(\Phi(x)) - \mu_0(\Phi(x))$ for all $x$;
**5**    Compute pairwise differences $\Delta\tau_{xz} = |\hat{\tau}(x) - \hat{\tau}(z)|$ for all $x, z$;
**6**    Set similarity threshold $\tau_{\text{thr}}$ (e.g., 20th percentile of $\Delta\tau_{xz}$);
**7**    Define positive set $\mathcal{P}_{\text{pos}} = \{(x, z) : \Delta\tau_{xz} \leq \tau_{\text{thr}}\}$;
**8**    Define negative set $\mathcal{P}_{\text{neg}} = \mathcal{D} \times \mathcal{D} \setminus \mathcal{P}_{\text{pos}}$;
    // Labels are fixed until next epoch
    // Mini-batch Training Phase
**9**    **for** step$Gets$1 **to** $E_l$ **do**
**10**        Sample $k$ pairs from $\mathcal{P}_{\text{pos}}$ and $k$ pairs from $\mathcal{P}_{\text{neg}}$ to build labeled batch $\mathcal{M}_l$;
**11**        Compute branch losses $\mathcal{L}_{\text{branch}_x}$ and $\mathcal{L}_{\text{branch}_z}$;
**12**        Compute contrastive loss

$$\mathcal{L}_{ctr}(\Phi(x), \Phi(z)) = \begin{cases} \|\Phi(x) - \Phi(z)\|_2^2, & (x, z) \in \mathcal{P}_{\text{pos}}, \\ \max\{0, \, m - \|\Phi(x) - \Phi(z)\|_2\}, & (x, z) \in \mathcal{P}_{\text{neg}}. \end{cases}$$

        Compute total loss $\mathcal{L}_{\text{tot}} = \mathcal{L}_{\text{branch}_x} + \mathcal{L}_{\text{branch}_z} + \lambda \sum_{(x,z) \in \mathcal{M}_l} \mathcal{L}_{ctr}(\Phi(x), \Phi(z))$;
**13**        Backpropagate $\mathcal{L}_{\text{tot}}$ and update model weights;

---

individuals with similar treatment effects form locally dense neighborhoods. At the same time, negative pairs such as $a$ and $n$ (empty cross with orange border) are pushed apart by at least the predefined margin $m$, preventing spurious proximity between individuals whose treatment effects are dissimilar. As a result, counterfactual estimates are obtained via interpolation within causally coherent neighborhoods, significantly mitigating the risk of unreliable extrapolation.

### 3.3 Implementation

The practical implementation of our model `HERMES` is described by the Algorithm 1.

The training alternates two phases: a *pairing update phase*, in which input pairs are dynamically labeled according to the current ITE predictions, and a *minibatch training phase*, in which the network is optimized using both factual losses and a contrastive regularizer. At the beginning of each epoch, the network computes current estimates of the ITE $\hat{\tau}_x$ for all samples. For every pair $(x, z)$, the absolute difference $\Delta\tau_{xz}$ is calculated, providing a measure of how similar their predicted treatment effects are. A similarity threshold $\tau_{\text{thr}}$ is then chosen, typically as a low quantile (e.g., the 20th percentile) of the distribution of $\Delta\tau_{xz}$. Pairs whose treatment effects differ less than this threshold are collected into the positive set $\mathcal{P}_{\text{pos}}$, while the remaining pairs form $\mathcal{P}_{\text{neg}}$.

Within each epoch, the optimization proceeds over mini-batches. For every mini-batch, a balanced number of positive and negative pairs is sampled to construct a labeled batch $\mathcal{M}_l$. Each element of the batch is processed through the two twin branches of the network, which share parameters and return potential outcome predictions and propensity scores for each individual. The branch-specific losses $\mathcal{L}_{\text{branch}_x}$ and $\mathcal{L}_{\text{branch}_z}$ are computed on the corresponding samples, combining squared error, cross-entropy on the propensity score, balancing penalties, and regularizers as defined in Equation equation 4. In parallel, the contrastive loss $\mathcal{L}_{ctr}$ is computed on the sampled pairs. For positive pairs, the embeddings are encouraged to be close in the latent space by minimizing their squared Euclidean distance $\|\Phi(x) - \Phi(z)\|_2^2$. For negative pairs, the model enforces a margin separation: if two embeddings are closer than a predefined margin $m$, a penalty proportional to the gap $(m - \|\Phi(x) - \Phi(z)\|_2)$ is applied, otherwise no penalty is incurred. This mechanism enforces a semantic geometry in the latent space where distance reflects difference in treatment effect values.

The total loss for a mini-batch $\mathcal{L}_{\text{tot}}$ is backpropagated, updating both the outcome heads and the representation encoder $\Phi(\cdot)$. By construction, the updates simultaneously reduce factual prediction error and enforce

local alignment of treated and control individuals with similar ITEs, which decreases the effective IPM in the relevant regions of the latent space.

A vulnerability of dynamically pairing samples based on self-estimated ITEs is the risk of a degenerate feedback loop during the early stages of training. If the initial ITE estimates ($\hat{\tau}_x$) are highly noisy or biased, the model might construct erroneous positive and negative pairs. To mitigate this "cold start" problem and ensure stability, HERMES implements several stabilizing mechanisms in its training process:

1. *factual warm-up phase:* the training begins with a warm-up phase driven exclusively by the factual branch losses $\mathcal{L}_{\text{branch}}$; the contrastive term $\mathcal{L}_{ctr}$ is disabled until the outcome heads have learned a grounded baseline representation; the purpose is to allow the outcome heads to learn a sufficiently reliable representation before pair construction begins to influence the latent space (activating it too early would expose training to unstable pair assignments); delaying $\mathcal{L}_{ctr}$ makes the selected pairs more meaningful by introducing the contrastive signal only after a factual understanding of the data.

2. *periodic labeling:* the pair labels are not updated at each gradient step, but periodically to prevent the latent geometry from oscillating wildly due to rapid changes in the pseudo-ITE estimates.

3. *persistent factual anchoring:* the factual loss remains fully active throughout the training process acting as a persistent anchor; it guarantees that, regardless of the contrastive pushes, the representations must always retain the necessary information to accurately predict the observed outcomes.

To empirically validate the resilience of HERMES against specific *failure modes*, i.e., extreme observational settings characterized by virtually zero overlap or severe unmeasured confounding, we conducted a stress test, demonstrating that the persistent factual anchor allows the model to self-correct and converge even when early pseudo-ITE estimates are catastrophically corrupted by massive artificial noise (see Appendix B.3).

Furthermore, HERMES does not require well-formed or reliable ITE-based clusters at the beginning of training. No such structure is assumed to be available *a priori*. During the warm-up phase, the model is optimized only through factual supervision, so the representation is not yet shaped by contrastive pair assignments. The purpose is to obtain preliminary ITE estimates that, although still imperfect, already contain a weak relative signal sufficient to initialize pair construction, without assuming accurate effect-homogeneous clusters from the outset. As training proceeds, these pairwise relations become progressively more reliable, and the local structure of the representation space is progressively refined rather than imposed in advance.

In summary, the algorithm alternates between updating pair labels based on the model's evolving ITE predictions and optimizing the network with a combination of factual and contrastive losses. The dynamic relabeling ensures that the latent space progressively adapts to the current predictive ability of the model, while the contrastive term structures the representation so that individuals with similar causal responses remain close, and those with very different responses are separated by at least the margin $m$. This self-supervised mechanism transforms the extrapolation problem of counterfactual prediction into a more stable interpolation problem, which is particularly advantageous in regions of limited overlap.

## 4    Applications to Benchmark and Real-World Datasets

We evaluate the proposed method on two standard benchmark datasets for CATE estimation, i.e., the *Infant Health and Development Program* (IHDP) proposed by Ramey et al. (1992) and the Jobs (Smith & Todd, 2005) datasets, by following the exact experimental methodology proposed by Shalit et al. (2017), and Johansson (2025) to ensure direct comparability with prior work.

We recall that, IHDP is a semi-synthetic benchmark dataset based on a real-world randomized controlled trial. To introduce confounding, the dataset has been modified by removing a non-random subset of the treated group, resulting in significant selection bias. The benchmark consists of 1,000 replications with simulated outcomes, allowing for robust evaluation using the known ground-truth ITEs. Each replication contains $N = 747$ units described by 25 covariates.

Jobs is based on the *National Supported Work* (NSW) program, a randomized job training experiment (LaLonde, 1986). To create a challenging observational setting with severe covariate shift, the original control group is replaced with a much larger, non-randomized control group from the *Panel Study of Income Dynamics* (PSID), following the setup by Dehejia & Wahba (2002). The final dataset contains $N = 3,084$ individuals (297 treated, 2,787 controls) described by 8 covariates.

### 4.1 Evaluation criteria

We categorize our metrics into three groups: *(i)* oracle metrics for estimation, *(ii)* diagnostic metrics for latent space analysis, and *(iii)* overlap and covariate-balance diagnostics related to the identifying assumptions discussed in Section 2.

### 4.2 Estimation accuracy

These metrics quantify the error of the model in predicting the treatment effect (Table 2).

*Root Mean Square Error* (RMSE) between the true ITE, $\tau(\mathbf{x})$, and the estimated CATE, $\hat{\tau}(\mathbf{x})$, over the $N_{\text{test}}$ units in the test set:

$$\text{RMSE} = \sqrt{\frac{1}{N_{\text{test}}} \sum_{i=1}^{N_{\text{test}}} (\hat{\tau}(\mathbf{x}_i) - \tau(\mathbf{x}_i))^2} \tag{6}$$

In our experiments, we use this metric on IHDP dataset.

*Absolute Error on Average Treatment Effect (ATE)* evaluates the model's bias by comparing the estimated ATE with the true ATE. This is crucial for assessing performance on policy-level decisions.

$$\epsilon_{\text{ATE}} = \left| \left( \frac{1}{N_{\text{test}}} \sum_{i=1}^{N_{\text{test}}} \hat{\tau}(\mathbf{x}_i) \right) - \left( \frac{1}{N_{\text{test}}} \sum_{i=1}^{N_{\text{test}}} \tau(\mathbf{x}_i) \right) \right| \tag{7}$$

In our experiments, this metric was also used on the IHDP dataset.

The *Absolute Error on Average Treatment Effect on the Treated (ATT)* is a metric which quantifies the model's bias on the *treated* subpopulation in terms of absolute difference between *(i)* the mean estimated CATE over the treated individuals and *(ii)* the ground-truth ATT, thus assessing how well the method recovers effects for those individuals which actually receive the treatment.

$$\epsilon_{\text{ATT}} = \left| \left( \frac{1}{N_{T=1}} \sum_{i:t_i=1} \hat{\tau}(\mathbf{x}_i) \right) - \text{ATT}_{\text{true}} \right| \tag{8}$$

In our experiments, we use this metric on the Jobs dataset.

The *Policy Risk ($R_{pol}$)* evaluates the utility of the learned CATE function, by quantifying the expected outcome if a treatment-assignment policy $\pi(\mathbf{x})$ was deployed. It is defined as the negative of the expected potential outcome under this policy, where a lower value indicates a better policy.

$$R_{\text{POL}}(\pi) = -\mathbb{E}\left[Y(1) \cdot \pi(\mathbf{x}) + Y(0) \cdot (1 - \pi(\mathbf{x}))\right] \tag{9}$$

Particularly relevant for Jobs, as it directly measures the true value of the model's recommendations.

On the semi-synthetic IHDP dataset, for each individual we have both $Y(0)$ and $Y(1)$, and therefore the calculation of these metrics was immediate. On the real Jobs dataset, where for each individual we have only one of these potential outcomes, we had to use a different methodology. As for the ATT, as suggested by (Shalit et al., 2017), Jobs contains a randomized population from the *National Supported Work* (NSW) experiment. The presence of the randomized subgroup gives a way to estimate the "ground truth" causal effect. Each subject is annotated with an indicator $E_i \in \{0, 1\}$, where $E_i = 1$ indicates membership in the

randomized population. We limit our causal evaluation to this subset, since for $E = 1$ the treatment assignment is randomized, which means that ignorability and overlap are valid by design. Consequently, causal quantities are identifiable from the observed results without requiring additional assumptions or modeling corrections. In the NSW subset, the ground truth ATT is directly identifiable as the difference between the treated and control groups. As for the Policy Risk, instead, it is estimated by applying the HERMES-induced policy to the NSW population and evaluating it by using a Horvitz–Thompson-type estimator, which uses only observed outcomes, without ever requiring counterfactuals. More specifically, let $\pi(\mathbf{x}_i)$ be the HERMES deterministic treatment policy, and let $t_i \in \{0, 1\}$ and $y_i$ be the observed treatment outcome for unit $i$. We define the treatment distribution as follows:

$$\hat{p}_1 = \frac{1}{n} \sum_{i=1}^{n} \pi(\mathbf{x}_i), \qquad \hat{p}_0 = 1 - \hat{p}_1,$$

and the empirical means of $Y$ among units whose observed treatment is consistent with the policy:

$$\bar{y}_{S_1} = \frac{\sum_{i=1}^{n} \pi(\mathbf{x}_i)\, t_i\, y_i}{\sum_{i=1}^{n} \pi(\mathbf{x}_i)\, t_i}, \qquad \bar{y}_{S_0} = \frac{\sum_{i=1}^{n} (1 - \pi(\mathbf{x}_i))(1 - t_i)\, y_i}{\sum_{i=1}^{n} (1 - \pi(\mathbf{x}_i))(1 - t_i)}.$$

Then, the estimated policy value is $\widehat{V}(\pi) = \hat{p}_1\, \bar{y}_{S_1} + \hat{p}_0\, \bar{y}_{S_0}$, and we define the policy risk as:

$$R_{\mathrm{pol}}(\pi) = 1 - \widehat{V}(\pi)$$

### 4.3 Diagnostic Metrics for Latent Space Analysis

These metrics have been designed to validate our central hypothesis: the contrastive regularizer proposed in this work successfully imparts a causally meaningful geometry to the latent space. (Table 5).

The *Maximum Mean Discrepancy (MMD)* measures the distributional discrepancy between the latent representations of the treated and control groups. In our experiments, we use two variants:

- *Linear-MMD: global balance* measure based on a linear kernel which computes the distance between the centroids of the treated and control distributions.

- *RBF-MMD: Gaussian Radial Basis Function* kernel, which is a kernel sensitive to all differences in distributions, including higher-order moments and *local micro-structures*; a low value indicates a fine-grained overlap, which is the direct target of our contrastive loss.

The *Semantic Alignment* is used to measure the Spearman's rank correlation $\rho$ between the Euclidean distances $\|\Phi(x_i) - \Phi(x_j)\|_2$ in the latent space and the absolute differences $|\hat{\tau}(x_i) - \hat{\tau}(x_j)|$ in the estimated treatment effects. A $\rho$ value close to 1 indicates that the loss has successfully organized the space such that geometric distance is semantically correlated to the treatment effect.

### 4.4 Metrics for Causal Assumption Verification

Finally, to assess whether the learned representation is consistent with the overlap and covariate-balance conditions typically sought in observational causal inference, we report the following diagnostics.

*Tail IPM* is a variant of IPM that focuses on the distributions tails, making it a sensitive measure for detecting violations of the *positivity* assumption in low-density regions of the latent space.

*k-NN overlap* is an empirical measure of *positivity*: for each point of the latent space, it calculates the proportion of its $k$-nearest neighbors that belong to the opposite treatment group. A high average value (e.g., $\geq 0.8$) indicates sufficient local overlap.

The *worst-case Standardized Mean Difference (worst-SMD)* is used as a post-weighting covariate-balance diagnostic on the observed covariates. After applying balancing weights derived from the model, it computes the *Standardized Mean Difference* (SMD) for each covariate. A worst-SMD value below the common threshold of 0.1 indicates improved comparability between treated and control groups with respect to measured pre-treatment variables.

### 4.5 Local Support and Extrapolation Diagnostics

While global balance and semantic alignment provide important evidence on the structure of the learned representation, they do not fully capture how the model behaves in regions where the overlap between treated and control units is limited. Since reliable counterfactual estimation critically depends on local support, we introduce a set of diagnostics specifically designed to evaluate the severity of extrapolation in challenging regions of the latent space, i.e regions where opposite-treatment neighbors are sparse or absent (Table 6).

*Nearest Opposite-Treatment Distance (NN distance).* For each unit, we compute the Euclidean distance in latent space to its closest neighbor belonging to the opposite treatment group. This metric directly quantifies the availability of local counterfactuals: lower values indicate that each unit can rely on nearby counterparts, reducing the need for extrapolation.

*Support-Gap Ratio.* We partition the latent space into regions of high and low support based on local overlap statistics. The support-gap ratio is defined as the ratio between the estimation error (e.g., RMSE or factual loss proxy) in low-support regions and the corresponding error in high-support regions. Values close to 1 indicate that performance degrades only marginally when overlap deteriorates.

*Hard-Gap Ratio.* To further test the representation, we define a subset of particularly challenging regions (e.g., bottom quantile of overlap or largest NN distances). The hard-gap ratio measures the error inflation in these regions relative to the rest of the space. Lower values indicate improved robustness in the most difficult parts of the latent space.

Together, these metrics provide a complementary perspective to standard balance diagnostics, explicitly quantifying how the learned representation mitigates extrapolation in regions where causal estimation is intrinsically more challenging.

### 4.6 Experimental Details

The following procedure was designed to ensure reproducibility and fair comparison. Source code in `PyTorch` is available online.[1] We performed a rigorous hyperparameter optimization (HPO) for each dataset separately using the *Optuna* framework [2] with a *Tree-structured Parzen Estimator* sampler. Each HPO process was run for 100 trials with different goals for the two benchmark datasets: *(i)* as for `IHDP`, the objective was to minimize the out-of-sample RMSE, averaged over all 1,000 replications, while *(ii)* as for `Jobs`, where the ground-truth ITE is unknown, the objective was to minimize the out-of-sample ATT Error, a standard practice for this benchmark.

The search spaces and the resulting optimal hyperparameter configurations for both datasets are summarized in Table 1. The analysis revealed a stable performance region for the key hyperparameters (margin $m$ and contrastive weight $\lambda_{\mathrm{ctr}}$), confirming the robustness of the selected configurations.

Table 1: Hyperparameter optimization search space and optimal values found for IHDP and Jobs.

| Hyperparameter | IHDP | | Jobs | |
|---|---|---|---|---|
| | Search Space | Optimal Value | Search Space | Optimal Value |
| *Learning Rate* (*lr*) | $[10^{-5}, 10^{-2}]$ | $3.19 \times 10^{-4}$ | $[10^{-4}, 10^{-3}]$ | $1.05 \times 10^{-4}$ |
| *Batch Size* | $\{32, 64\}$ | 32 | $\{32, 64\}$ | 32 |
| *Margin* (*m*) | $[0.0, 1.0]$ | 0.432 | $[0.2, 0.8]$ | 0.561 |
| *Contrastive Weight* ($\lambda_{\mathrm{ctr}}$) | $[0.1, 1.0]$ | 1.0 | $[0.01, 0.2]$ | 0.145 |
| *Pair Percentage* (*perc*) | − | − | $[10, 50]$ | 17 |

Among all the hyper-parameter configurations, the one achieving best results is the following:

- *Training schedule:* 500 epochs, batch size 32, Adam optimizer, gradient clipping at $\ell_2$-norm 2.0, no AMP. For stability and efficiency, We use a Reduce-on-Plateau scheduler, and early stopping on RMSE values on validation set, validation split 0.2.

---

[1] https://anonymous.4open.science/r/TMLR20260119A1042HERMES/README.md
[2] https://optuna.org/

- *Pair mining:* relabeling every $K=3$ epochs, warm-up 20 epochs for $\mathcal{L}_{branch}$, L2 regularization weight $\lambda_{\mathrm{reg}}=0.1$, not using $\mathcal{L}_{bce}$ in the loss $\mathcal{L}_{branch}$

Beyond the parameters tuned via *Optuna*, all remaining architectural and loss settings were kept fixed to ensure methodological comparability with the original `BCAUSS` implementation. Specifically, regarding equation 4, we maintained the balancing term at $\lambda_{\mathrm{bal}} = 1$ to provide a stable trade-off between factual fitting and covariate rebalancing. Conversely, the targeted regularization ($\lambda_{\mathrm{targ}}$) and the binary cross-entropy supervision ($\lambda_{\mathrm{bce}}$) were disabled ($\lambda = 0$), as preliminary tests showed no consistent benefits when combined with our contrastive regularizer.

## 5 Results

Table 2: On the left performance comparison on IHDP (out-of-sample). RMSE and ATE error $\epsilon_{\mathrm{ATE}}$ as mean $\pm$ standard deviation. On the right performance comparison on the Jobs benchmark (out-of-sample). Best results are in bold.

| | IHDP | | Jobs | |
|---|---|---|---|---|
| **Model** | RMSE | $\epsilon_{\mathrm{ATE}}$ | $\epsilon_{\mathrm{ATT}}$ | $R_{\mathrm{POL}}$ |
| `TARNet` (Shalit et al., 2017) | $0.95 \pm .01$ | $0.28 \pm .01$ | $0.05 \pm 0.02$ | $0.17 \pm 0.0$ |
| `CFR-MMD` (Shalit et al., 2017) | $0.78 \pm .01$ | $0.31 \pm .01$ | $0.04 \pm 0.01$ | $0.18 \pm 0.0$ |
| `CFR-WAS` (Shalit et al., 2017) | $0.76 \pm .01$ | $0.27 \pm .01$ | $0.05 \pm 0.01$ | $0.17 \pm 0.0$ |
| `DragonNet` (Shi et al., 2019) | – | $0.20 \pm .01$ | – | – |
| `BCAUSS` (Tesei et al., 2023) | – | $0.15 \pm .01$ | $0.05 \pm 0.02$ | – |
| `HERMES` | $\mathbf{0.54 \pm 0.1}$ | $\mathbf{0.13 \pm 0.1}$ | $\mathbf{0.05 \pm 0.02}$ | $\mathbf{0.08 \pm 0.01}$ |

### 5.1 Comparison on IHDP and on Jobs

To evaluate the effectiveness of `HERMES`, we compared its performance on `IHDP` with that obtained by *(i)* classical methods proposed in Shalit et al. (2017), i.e., `TARNet`, `CFR-MMD`, and `CFR-WAS`, *(ii)* `DragonNet` (Shi et al., 2019), and *(iii)* the most recent state-of-the-art `BCAUSS` (Tesei et al., 2023). As shown in Table 2, `HERMES` outperforms all the methods in terms of RMSE, achieving a value of **0.54**, which improves the result of the best baseline `CFR-Was` (0.76). In terms of ATE, `HERMES` (**0.13**) is more accurate than all the original benchmark models, including `CFR-Was` (ATE 0.27). In conclusion, `HERMES` is extremely competitive in estimating the mean effect. Furthermore, with an ATE value of **0.13**, our approach reduces the population-level bias even compared to the best `BCAUSS` configuration (0.15).

As regards the `Jobs` dataset, as shown in Table 2, while `HERMES` achieves an ATT that is competitive with the state-of-the-art baselines, its advantage emerges in the *Policy Risk*. Indeed, it achieves an $R_{\mathrm{pol}}$ of **0.08 $\pm$ 0.01**, which is significant reduction compared to the best-performing baselines. This finding suggests that while the point-estimate accuracy on the treated group is similar, the CATE function learned by `HERMES` leads to better and more reliable treatment decisions. The geometric latent structure induced by our local, effect-driven regularizer appears to generate more robust policy recommendations, which is a key desideratum for deploying CATE models in real-world decision-making contexts.

## 5.2 Ablation Study

To better understand which components are responsible for the empirical performance of HERMES, we analyze the method along two complementary axes: *(i) structural ablation*, aimed at verifying the necessity of its core architectural components; and *(ii) optimization and sensitivity analysis*, aimed at evaluating the robustness of the training procedure and of the main hyperparameters in Algorithm 1.

Table 3 reports the ablation results on the IHDP benchmark and is organized into two parts. *Structural ablation* isolates the contribution of the structural components of the framework, namely the contrastive objective, the pair-construction rule, and the dynamic relabeling mechanism. *Optimization and sensitivity analysis* instead evaluates the role of the training schedule and the sensitivity of the model.

A clear pattern emerges from *Structural ablation*: the best result is only achieved when HERMES combines all its parts, namely, contrastive regularization, ITE-based pairing, and dynamic relabeling. Removing the contrastive term reduces performance, demonstrating that the improvement is not solely due to the shared backbone. Replacing dynamic relabeling with static ITE-based pairs results in the most significant deterioration among the Siamese variants, indicating that static pseudo-labels aren't useful and that regular pair updates are essential to the method. Random and covariate-based pairing also underperform compared to the full model, confirming that the improvement is not solely due to the Siamese architecture, but rather to the organisation of the latent space according to treatment-effect similarity.

*Optimization and sensitivity analysis* shows that the effectiveness of HERMES also depends on how contrastive supervision is introduced during training. In particular, removing the warm-up phase, or reducing it to only 5 epochs, increases RMSE from 0.540 to about 0.588–0.589, suggesting that the model requires a sufficiently stable representation before the contrastive signal becomes fully reliable. An even stronger degradation is observed when pseudo-ITE refresh is disabled altogether, confirming that dynamic relabeling is not only conceptually, but also practically necessary for stable optimization. By contrast, refreshing pair assignments every 5 epochs remains close to the reference model, which suggests that the method is robust to moderate changes in refresh frequency and allows some computational flexibility.

The sensitivity analysis further shows that HERMES remains reasonably stable around its default configuration. Moderate changes in the contrastive weight and in the number of sampled pairs per step do not substantially alter performance, whereas more visible degradations appear when the margin is reduced or when the threshold is moved away from its default value.

Overall, these results indicate that the success of HERMES is due to the combination of a dynamic effect-aware pairing and a training schedule that keeps the contrastive supervision stable throughout optimization.

We replicate the ablation study described above on JOBS (see Table 4). Since oracle individual treatment effects are not available as in IHDP, we report $\epsilon_{\text{ATT}}$ and policy risk ($R_{\text{POL}}$) instead.

Overall, the JOBS results confirm that the contrastive component remains important, although the picture is more nuanced than on IHDP because the two evaluation metrics emphasize different aspects of performance. In *Structural ablation*, removing the contrastive term slightly worsens $\epsilon_{\text{ATT}}$ and leads to a more substantial degradation in policy risk, indicating that the contrastive geometry is particularly beneficial for downstream treatment decisions. The full dynamic ITE-based configuration achieves the best policy-risk value among the structural variants, whereas the BCAUSS backbone attains a lower ATT error but at the cost of a clearly worse policy risk. This suggests that, on JOBS, the main gain of HERMES lies less in improving pointwise estimation on the treated subgroup and more in learning a representation that supports better treatment-allocation policies.

*Optimization and sensitivity analysis* shows that the optimization schedule is also important on JOBS. Removing the warm-up phase degrades both metrics, confirming that contrastive supervision should not be activated too early. Disabling pair refresh also harms performance, whereas refreshing every 5 epochs improves ATT but worsens policy risk, suggesting a trade-off between local effect fitting and policy robustness. The analysis further indicate that the model is reasonably stable, although some variants can improve one metric while degrading the other. In particular, higher contrastive weight slightly improves both ATT and

Table 3: Unified ablation study on IHDP. *Structural ablation* reports the contribution of the structural components of `HERMES`, namely the contrastive term, the pair-construction rule, and the dynamic relabeling mechanism. *Optimization and sensitivity analysis* evaluates optimization-related choices and the sensitivity of the method around the reference configuration.

| Variant | Ctr. | Pair criterion | Dyn. | Warm-up | Refresh | $\lambda$ | $m$ | Thr. | $k$ | RMSE / $\epsilon_{\text{ATE}}$ |
|---|---|---|---|---|---|---|---|---|---|---|
| *Structural ablation* | | | | | | | | | | |
| BCAUSS backbone | No | – | No | – | – | 0.0 | – | – | – | 0.809 / 0.15 |
| No contrastive | No | – | No | 20 epochs | every 3 ep. | 0.0 | 0.432 | 20% | 32 | 0.554 / 0.138 |
| Random pairs | Yes | Random | No | 20 epochs | every 3 ep. | 1.0 | 0.432 | 20% | 32 | 0.557 / 0.135 |
| Covariate pairs | Yes | Covariate similarity | No | 20 epochs | every 3 ep. | 1.0 | 0.432 | 20% | 32 | 0.556 / 0.140 |
| Static ITE pairs | Yes | ITE similarity | No | 20 epochs | fixed | 1.0 | 0.432 | 20% | 32 | 0.596 / 0.140 |
| Dynamic ITE pairs | Yes | ITE similarity | Yes | 20 epochs | every 3 ep. | 1.0 | 0.432 | 20% | 32 | **0.540 / 0.130** |
| *Optimization and sensitivity analysis* | | | | | | | | | | |
| HERMES_Full | Yes | ITE similarity | Yes | 20 epochs | every 3 ep. | 1.0 | 0.432 | 20% | 32 | **0.540 / 0.130** |
| HERMES_NoWarmup | Yes | ITE similarity | Yes | 0 epochs | every 3 ep. | 1.0 | 0.432 | 20% | 32 | 0.588 / 0.159 |
| HERMES_Warmup_5 | Yes | ITE similarity | Yes | 5 epochs | every 3 ep. | 1.0 | 0.432 | 20% | 32 | 0.589 / 0.169 |
| HERMES_RefreshEvery5 | Yes | ITE similarity | Yes | 20 epochs | every 5 ep. | 1.0 | 0.432 | 20% | 32 | 0.547 / 0.127 |
| Low contrastive weight | Yes | ITE similarity | Yes | 20 epochs | every 3 ep. | 0.1 | 0.432 | 20% | 32 | 0.578 / 0.170 |
| High contrastive weight | Yes | ITE similarity | Yes | 20 epochs | every 3 ep. | 2.0 | 0.432 | 20% | 32 | 0.543 / 0.127 |
| Small margin | Yes | ITE similarity | Yes | 20 epochs | every 3 ep. | 1.0 | 0.200 | 20% | 32 | 0.556 / 0.153 |
| Large margin | Yes | ITE similarity | Yes | 20 epochs | every 3 ep. | 1.0 | 1.000 | 20% | 32 | 0.550 / 0.125 |
| Tighter threshold | Yes | ITE similarity | Yes | 20 epochs | every 3 ep. | 1.0 | 0.432 | 10% | 32 | 0.547 / 0.156 |
| Looser threshold | Yes | ITE similarity | Yes | 20 epochs | every 3 ep. | 1.0 | 0.432 | 30% | 32 | 0.556 / 0.144 |
| Fewer pairs per step | Yes | ITE similarity | Yes | 20 epochs | every 3 ep. | 1.0 | 0.432 | 20% | 16 | 0.560 / 0.142 |
| More pairs per step | Yes | ITE similarity | Yes | 20 epochs | every 3 ep. | 1.0 | 0.432 | 20% | 64 | 0.553 / 0.137 |

policy risk, tighter thresholds improve ATT at the expense of policy risk, and using more pairs per step slightly improves policy risk but worsens ATT.

These results support the same qualitative conclusion as on IHDP: the effectiveness of `HERMES` does not come from the Siamese backbone alone, but from the interaction between effect-aware pairing, contrastive regularization, and dynamic relabeling. On JOBS, however, this benefit is expressed most clearly in policy quality rather than uniformly across all estimation metrics.

## 5.3 Latent Space Analysis

The results in Tables 2 show that `HERMES` is highly competitive, when compared to other state of the art procedures. To understand the reasons for this performance, we conducted a diagnostic analysis of the latent space on the IHDP dataset and compared it to the `BCAUSS` latent space, measuring treated–control distributional discrepancies and examining the geometry via low-dimensional projections.

Considering together the seven metrics reported in Table 5, a coherent picture emerges of the benefits introduced by the `HERMES` compared to `BCAUSS`. First of all, the *factual MSE* is reduced by 6%, confirming that the architecture improves the predictive component, the first component of the Shalit bound. On the balancing side, the *Linear-MMD* highlights a convergence of the centroids (-8%), while the more sensitive *RBF-MMD* drops by 13%, indicating a much closer overlap of the local structures; coherently, the *Tail-IPM* decreases by 12%, a sign that even the low overlap regions are corrected. These improvements do not erode the positivity: the *k-NN overlap* remains stable above 0.80, the commonly accepted threshold. In parallel, the latent geometry becomes more semantic: the Spearman $\rho$ between latent distance and ITE gap goes from 0.80 to 0.92, while the *worst-SMD* drops from 0.060 to 0.028, well below the 0.10 limit indicated for ignorability. Overall, `HERMES` simultaneously reduces the predictive error and the causal divergence, preserving the overlap: the theoretical bound tightens by about 45% and the consequent decrease of RMSE and ATE-bias empirically confirms the better empirical performance together with improved overlap and balance diagnostics. The contrastive training of `HERMES` simultaneously lowers the observed error and the

Table 4: Unified ablation study on JOBS. *Structural ablation* reports the contribution of the structural components of `HERMES`, namely the contrastive term and the dynamic ITE-based pairing strategy. *Optimization and sensitivity analysis* evaluates optimization-related choices and the sensitivity of the method around the reference configuration.

| Variant | Ctr. | Pair criterion | Dyn. | Warm-up | Refresh | $\lambda$ | $m$ | Thr. | $k$ | $\epsilon_{\mathbf{ATT}}$ / $R_{\mathbf{POL}}$ |
|---|---|---|---|---|---|---|---|---|---|---|
| *Structural ablation* | | | | | | | | | | |
| BCAUSS backbone | No | – | No | – | – | 0.0 | – | – | – | 0.05 / 0.086 |
| No contrastive | No | – | No | 20 epochs | every 3 ep. | 0.145 | 0.561 | 17% | 32 | 0.06 / 0.085 |
| Random pairs | Yes | Random | No | 20 epochs | every 3 ep. | 0.145 | 0.561 | 17% | 32 | 0.061 / 0.09 |
| Covariate pairs | Yes | Covariate similarity | No | 20 epochs | every 3 ep. | 0.145 | 0.561 | 17% | 32 | 0.056 / 0.0820 |
| Static ITE pairs | Yes | ITE similarity | No | 20 epochs | fixed | 0.145 | 0.561 | 17% | 32 | 0.0596 / 0.083 |
| Dynamic ITE pairs | Yes | ITE similarity | Yes | 20 epochs | every 3 ep. | 0.145 | 0.561 | 17% | 32 | **0.050 / 0.080** |
| *Optimization and sensitivity analysis* | | | | | | | | | | |
| HERMES_Full | Yes | ITE similarity | Yes | 20 epochs | every 3 ep. | 0.145 | 0.561 | 17% | 32 | **0.050 / 0.080** |
| HERMES_NoWarmup | Yes | ITE similarity | Yes | 0 epochs | every 3 ep. | 0.145 | 0.561 | 17% | 32 | 0.055 / 0.081 |
| HERMES_Warmup_5 | Yes | ITE similarity | Yes | 5 epochs | every 3 ep. | 0.145 | 0.561 | 17% | 32 | 0.049 / 0.081 |
| HERMES_RefreshEvery5 | Yes | ITE similarity | Yes | 20 epochs | every 5 ep. | 0.145 | 0.561 | 17% | 32 | 0.054 / 0.085 |
| Low contrastive weight | Yes | ITE similarity | Yes | 20 epochs | every 3 ep. | 0.010 | 0.561 | 17% | 32 | 0.055 / 0.082 |
| High contrastive weight | Yes | ITE similarity | Yes | 20 epochs | every 3 ep. | 0.200 | 0.561 | 17% | 32 | 0.057 / 0.080 |
| Small margin | Yes | ITE similarity | Yes | 20 epochs | every 3 ep. | 0.145 | 0.200 | 17% | 32 | 0.050 / 0.080 |
| Large margin | Yes | ITE similarity | Yes | 20 epochs | every 3 ep. | 0.145 | 0.800 | 17% | 32 | 0.050 / 0.080 |
| Tighter threshold | Yes | ITE similarity | Yes | 20 epochs | every 3 ep. | 0.145 | 0.561 | 10% | 32 | 0.053 / 0.082 |
| Looser threshold | Yes | ITE similarity | Yes | 20 epochs | every 3 ep. | 0.145 | 0.561 | 30% | 32 | 0.050 / 0.083 |
| Fewer pairs per step | Yes | ITE similarity | Yes | 20 epochs | every 3 ep. | 0.145 | 0.561 | 17% | 16 | 0.057 / 0.082 |
| More pairs per step | Yes | ITE similarity | Yes | 20 epochs | every 3 ep. | 0.145 | 0.561 | 17% | 64 | 0.055 / 0.081 |

Table 5: Comparison of key performance metrics between `BCAUSS` and `HERMES` on `IHDP`. The $\Delta$ % column indicates the percentage change. Bold values denote the superior result for each metric.

| Metric (mean $\pm$ std) | BCAUSS | HERMES | $\Delta$ % |
|---|---|---|---|
| *Factual MSE* | $0.138 \pm 0.015$ | $\mathbf{0.130 \pm 0.014}$ | $\downarrow 6$ % |
| *Linear MMD* | $0.0185 \pm 0.019$ | $\mathbf{0.0170 \pm 0.019}$ | $\downarrow 8$ % |
| *RBF MMD* | $0.0202 \pm 0.009$ | $\mathbf{0.0175 \pm 0.014}$ | $\downarrow 13$ % |
| *Tail IPM* | $1.53 \pm 0.23$ | $\mathbf{1.34 \pm 0.18}$ | $\downarrow 12$ % |
| *Spearman's $\rho$* | $0.796$ | $\mathbf{0.923}$ | $\uparrow 16$ % |
| *k-NN Overlap* | $0.861 \pm 0.023$ | $0.810 \pm 0.027$ | – |
| *worst-SMD (latent)* | $0.060$ (sim) | $\mathbf{0.028}$ (sim) | – |

treated/control dissimilarity, resulting in an embedding that is more consistent with overlap and observed-covariate balance diagnostics. As a result, the theoretical Shalit bound (see 2) on the generalization error is significantly reduced from $\approx 0.55$ to $\approx 0.30$. This reduction in the bound, equal to almost 45%, is reflected in a more accurate estimate of the RMSE, demonstrating the superiority of the new model for the estimation of individual causal effects.

Recall that the Shalit et al. bound is based on three basic hypotheses: *SUTVA*, *Conditional ignorability*, and *Positivity*. In our context, the first premise is guaranteed by the simulated nature of the `IHDP` (no patient influences the outcome of another, and the treatment is well specified). The other two are instead empirically verified in the latent space learned by `HERMES`:

- *Ignorability:* after a *inverse probability of treatment weighting* re-weighting, the 25 `IHDP` covariates show a median worst-SMD of 0.04; 95% of the replications remain below the 0.10 threshold proposed by Austin (2009), which is a plausible statistical control of confounders.

- *Positivity.* The k-NN-based overlap value is 0.887 for positive pairs and 0.857 for negative pairs, both higher than the commonly used value of 0.8, meaning that no latent region remains without a treated or control counterpart.

Under this diagnostic evidence, the reduction of the MMD-RBF ($-13$ %) translates into a decrease in the IPM term of the bound and, consequently, into an effective improvement of the MSE:

$$\text{MSE} \leq 2\big(\text{MSE}_{\text{fact}} + \text{IPM}_{\text{RBF}}\big) = 2(0.13 + 0.0175) = 0.295$$

which is numerically consistent with the empirical value $(0.54)^2 = 0.292$. In other words, HERMES not only respects the theoretical bound, but in practice achieves a gain $\sim 45$ %. The choice of adopting the MMD-RBF is strategic, since it combines in a single measure the theoretical, geometric and practical needs of our study. On the theoretical level, the Shalit bound is defined in terms of an IPM on 1-Lipschitz functions, and the MMD-RBF provides an upper bound that is both efficient (with convergence in $\mathcal{O}(n^{-1/2})$) and easy to estimate empirically. Consequently, reducing the MMD-RBF guarantees, by construction, to tighten the term that governs the generalization error of the MSE.

In addition to this formal guarantee, the RBF kernel offers a crucial advantage on the geometric level. Unlike the linear kernel, which simply compares centroids and can ignore differences in variance or shape, the Gaussian kernel is sensitive to local discrepancies. It gives more weight to the similarities between neighboring points, making it an ideal indicator precisely for those micro-asymmetries that our contrastive loss is designed to regularize. Finally, these advantages are combined with important practical properties: the MMD-RBF is differentiable, computationally efficient and stable, which makes it a reliable indicator both during training and evaluation.

Table 6 reports the results observed during the testing phase, which confirmed the trend of the training, with the exception of an increase of *Linear-MMD* ($+293\%$) which depends on the centers of mass of the two distributions which are pushed away by the margin effect. However, the greater variance introduced by unseen samples slightly shifts the centroids without compromising the local coherence, as confirmed by the simultaneous decrease of *MMD-RBF*, *Tail-IPM* and *worst-SMD*.

Table 6: Comparison of the main metrics on the IHDP test-set The column $\Delta$ % indicates the variation of HERMES with respect to BCAUSS. The best value per metric is in **bold**.

| Metric (mean $\pm$ std) | BCAUSS | HERMES | $\Delta$ % |
|---|---|---|---|
| *Linear MMD* | **0.215 $\pm$ 0.873** | 0.846 $\pm$ 2.517 | ↑ 293 % |
| *MMD-RBF* | 0.157 $\pm$ 0.091 | **0.115 $\pm$ 0.068** | ↓ 27 % |
| *Tail IPM* | 13.65 $\pm$ 13.94 | **10.77 $\pm$ 11.59** | ↓ 21 % |
| *Spearman's $\rho$* | 0.784 $\pm$ 0.072 | **0.918 $\pm$ 0.029** | ↑ 17 % |
| *k-NN Overlap* | 0.873 $\pm$ 0.022 | **0.881 $\pm$ 0.021** | ↑ 1 % |
| *worst-SMD (latent)* | 0.103 $\pm$ 0.040 | **0.054 $\pm$ 0.022** | ↓ 48 % |

While the global diagnostics reported above already indicate that HERMES improves the overall geometry of the latent space, they do not fully characterize what happens in the most challenging regions, namely where treated and control groups are weakly interleaved. To better assess this aspect, Table 7 reports three complementary local-support diagnostics on a held-out IHDP replication. The first quantity, the *nearest opposite-treatment distance*, measures how far an individual lies from its closest counterpart in the opposite treatment group. A smaller value indicates that counterfactual comparisons rely on nearby units rather than on distant extrapolation. Here, HERMES achieves a substantially lower value than BCAUSS (0.0225 vs. 0.0329), suggesting that its latent space provides closer cross-treatment neighbors. The *support-gap ratio* and the *hard-gap ratio* compare the estimation error in low-support or difficult regions against the error observed in easier parts of the latent space. In both cases, lower values indicate that the method degrades less severely when overlap deteriorates. HERMES improves over BCAUSS on both diagnostics, reducing the support-gap ratio from 1.155 to 1.028 and the hard-gap ratio from 1.317 to 1.059. Taken together, these results are consistent with the interpretation that the contrastive organization induced by HERMES makes the most problematic

regions of the representation space more amenable to local interpolation, thereby mitigating the practical severity of extrapolation. In other words, the gain is not only global: it also appears in those parts of the latent space where reliable counterfactual estimation is intrinsically more difficult.

Table 7: Local-support diagnostics on challenging regions of the latent space. Lower values indicate better local comparability between treated and control units and smaller error inflation in low-support regimes.

| Metric | BCAUSS | HERMES | Δ vs BCAUSS |
|---|---|---|---|
| *NN opposite distance* ↓ | 0.0329 | **0.0225** | $-31.7\%$ |
| *Support-gap ratio* ↓ | 1.155 | **1.028** | $-11.0\%$ |
| *Hard-gap ratio* ↓ | 1.317 | **1.059** | $-19.6\%$ |

Figure 3 provides two views of the *test* embeddings produced on `IHDP` . The PCA panel shows treated and control clouds with nearly identical spread but slightly shifted centroids, which is consistent with a non–zero *linear* MMD (centroid mismatch). In addition, the t-SNE displays an almost complete interleaving of the two groups along the manifold, confirming the result in Table 6.

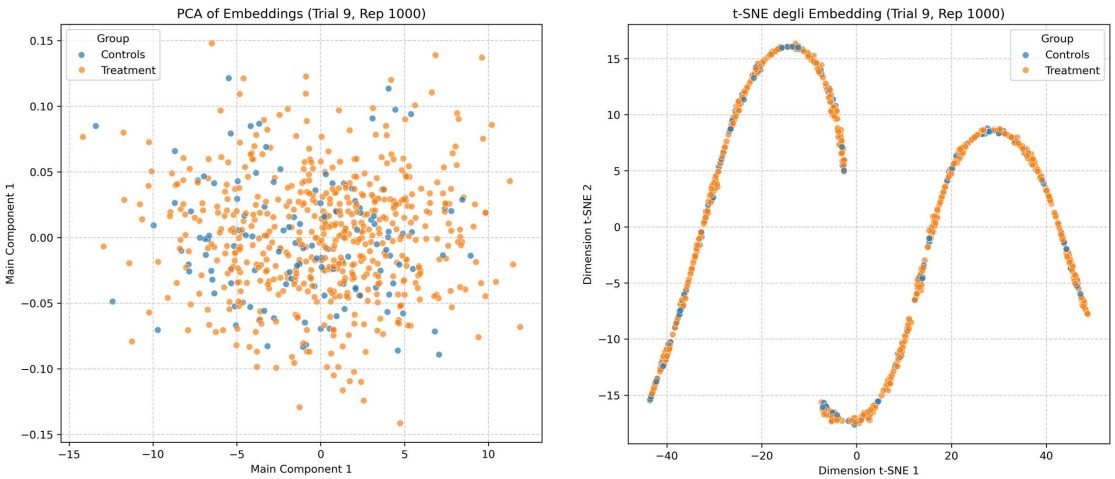

Figure 3: 2D projections of the latent embeddings onto the IHDP test set. *Left (PCA):* The distributions of treated (blue) and controls (orange) have a similar shape but slightly misaligned centroids, consistent with a non-zero linear MMD. *Right (t-SNE):* The projection reveals the almost perfect local overlap of the two groups along the learned manifold, explaining the excellent MMD value with RBF kernel and the low error on the ATE.

## 6   Conclusions

The use of a Siamese network with contrastive loss to estimate CATE introduces several practical and theoretical consequences that improve the quality of predictions and their interpretability.

Thanks to the explicit comparison between pairs of patients in training, the network learns latent representations in which subjects with similar treatment effects are close in the latent space, while those with marked differences are separated beyond an adaptive margin. This mechanism allows producing estimates of ITE personalized for each individual, rather than limiting oneself to a population mean value. In practice, for a new patient $x$, the estimated ITE $\hat{\tau}(x) = h_1(\Phi(x)) - h_0(\Phi(x))$ reflects the expected response by comparing it with the most "analogous" neighbors in terms of covariates and potential response. The "pairwise" organization of contrastive loss induces a geometric structure in the latent space that favors the formation of homogeneous clusters with respect to the treatment effect. This structure can be easily explored through dimensional reduction techniques (e.g. t-SNE), allowing to identify groups of subjects with

similar characteristics and responses. This facilitates the local interpretation of the results: a clinician or a policy-maker can visualize why two patients are considered "close" (same region of the latent space) and understand which covariates or latent differences determine similarities or divergences in the response to the treatment. The latent space generated by the model can be used to show the operator (clinician, economist, policy-maker) a group of patients "similar" to the one being examined. For example, by presenting a box of subjects with similar covariates and ITE, empirical evidence is provided for an *informed consent* process, encouraging a dialogue based on real and comparable cases. This function improves transparency and trust in therapeutic or policy choices because cases of success (or failure) in similar situations are highlighted. The geometric structure of the latent space allows monitoring any disparities in ITE between demographic subgroups (e.g. age, gender, ethnicity). By developing mean distance or overlap metrics between the latent distributions of each subgroup, it is possible to continuously audit and verify whether certain populations remain "out of balance" or have systematic biases in treatment estimates. If gaps emerge, hyperparameters of the contrastive loss (e.g. weight $\lambda$) or recalibrate the pair selection can be adjusted to dynamically reduce inequalities.

In summary, the adoption of a contrastive Siamese network for CATE estimation combines two main advantages: (i) an effective reduction of the IPM between latent distributions of treated and controls, resulting in a tighter theoretical bound on MSE, and (ii) greater interpretability and practical utility of the results, thanks to the structured organization of the representation space that supports clinical decision making and equity auditing.

## Broader Impact

While `HERMES` demonstrates significant improvements in the accuracy and interpretability of CATE estimation, its application in high-stakes domains, such as personalized medicine and public policy, requires careful ethical consideration. Like all causal inference methods relying on observational data, the validity of `HERMES`'s estimates depends on the assumption of unconfoundedness, which is fundamentally untestable. Since models can only learn from variables that are explicitly recorded, there is no purely statistical method to definitively prove the absence of unobserved factors that might simultaneously influence both the treatment assignment and the final outcome. The presence of such unmeasured confounders can lead to biased effect estimations, a risk that is particularly critical when human health or welfare is involved. Therefore, `HERMES` should be deployed strictly as a decision-support tool rather than an autonomous decision-maker. It is designed to assist, not replace, the nuanced judgment of medical professionals and policy-makers, ensuring human oversight to mitigate the risk of propagating automated biases in sensitive, real-world interventions.

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

# A  Proof

## A.1  Purpose and Scope of the Derivation

This section provides the formal proof of Theorem 3, which establishes an upper bound on $\text{EMSE}_\Phi$ based on the decomposition IPM. The goal is to prove that the contrastive regularization in `HERMES` reduces the effective IPM by minimizing local discrepancies within clusters of similar effects and controlling structural separation via the margin $m$.

## 1. Mixture Representation of Distributions

Let $\mathcal{R}$ be the representation (embedding) space induced by the function $\Phi : \mathcal{X} \to \mathcal{R}$. We assume that $\mathcal{R}$ admits a finite partition into $K$ disjoint subsets $\{C_k\}_{k=1}^K$, such that $\mathcal{R} = \bigcup_{k=1}^K C_k$. Each cluster $C_k$ represents a region of the latent space containing individuals with similar ITE.

To formalize this structure, we introduce a discrete latent variable $Z \in \{1, \ldots, K\}$ that assigns each embedding $r \in \mathcal{R}$ to its corresponding cluster. This allows us to express the global distributions of treated and control units as *mixtures* of simpler, cluster-specific distributions. In particular, the representation distributions for the treated and control groups can be written as follows:

$$p_\Phi^1(r) = \sum_{k=1}^K \pi_{1,k}\, p_\Phi^{1,k}(r), \qquad p_\Phi^0(r) = \sum_{k=1}^K \pi_{0,k}\, p_\Phi^{0,k}(r). \tag{10}$$

- $p_\Phi^{t,k}(r) = p(\Phi(X) = r \mid T = t, Z = k)$ denotes the *local* probability density of representations for treatment group $t$ within cluster $k$.

- $\pi_{t,k} = \mathbb{P}(Z = k \mid T = t)$ are the corresponding mixing coefficients, satisfying $\sum_{k=1}^K \pi_{t,k} = 1$ for each $t \in \{0,1\}$, which quantify the proportion of units from each treatment group whose representations fall into cluster $C_k$.

This mixture representation is purely probabilistic and introduces no approximation. Its role is to make explicit the multimodal structure of the latent space, which will allow us to decompose global distributional discrepancies into interpretable, cluster-level contributions in the following section.

## 2. Analytical Decomposition of the IPM Metric

Let $\mathcal{G}$ be a symmetric family of bounded real-valued functions. The global IPM is defined as:

$$\text{IPM}_{\mathcal{G}}(p_\Phi^1, p_\Phi^0) = \sup_{f \in \mathcal{G}} \left| \int_{\mathcal{R}} f(r)\, p_\Phi^1(r)\, dr - \int_{\mathcal{R}} f(r)\, p_\Phi^0(r)\, dr \right|. \tag{11}$$

Our goal is to express this *global* discrepancy in terms of *cluster-level* discrepancies induced by the mixture representation introduced above. We aim to decompose the global IPM into: *(i)* a weighted sum of *local IPMs* measuring the alignment between treated and control units within clusters of similar ITE, and *(ii)* a residual term capturing structural imbalances across clusters, which will be shown noting the influence of the contrastive margin $m$. Here are the steps of decomposition:

1. *Mixture substitution.* Using the mixture representations introduced above, we rewrite the global treated and control distributions as convex combinations of their cluster-conditional components. By substituting these expressions into the definition of the IPM yields:

$$\text{IPM}_{\mathcal{G}}(p_\Phi^1, p_\Phi^0) = \sup_{f \in \mathcal{G}} \left| \int_{\mathcal{R}} f(r) \left( \sum_{k=1}^K \pi_{1,k} p_\Phi^{1,k}(r) \right) dr - \int_{\mathcal{R}} f(r) \left( \sum_{k=1}^K \pi_{0,k} p_\Phi^{0,k}(r) \right) dr \right|$$

2. *Linearity of the integral.* We now make explicit the contribution of each mixture component by using properties of the integral. Since integration is a linear operator and the mixture weights $\pi_{t,k}$ are constants with respect to the integration variable $r$, we have:

$$\int_{\mathcal{R}} f(r) \left( \sum_{k=1}^{K} \pi_{1,k} p_{\Phi}^{1,k}(r) \right) dr = \sum_{k=1}^{K} \pi_{1,k} \int_{\mathcal{R}} f(r) \, p_{\Phi}^{1,k}(r) \, dr,$$

$$\int_{\mathcal{R}} f(r) \left( \sum_{k=1}^{K} \pi_{0,k} p_{\Phi}^{0,k}(r) \right) dr = \sum_{k=1}^{K} \pi_{0,k} \int_{\mathcal{R}} f(r) \, p_{\Phi}^{0,k}(r) \, dr.$$

By substituting these expressions into the definition of the IPM then we can write:

$$\mathrm{IPM}_{\mathcal{G}}(p_{\Phi}^1, p_{\Phi}^0) = \sup_{f \in \mathcal{G}} \left| \sum_{k=1}^{K} \pi_{1,k} \int_{\mathcal{R}} f(r) \, p_{\Phi}^{1,k}(r) \, dr - \sum_{k=1}^{K} \pi_{0,k} \int_{\mathcal{R}} f(r) \, p_{\Phi}^{0,k}(r) \, dr \right|$$

3. *Expected value notation.* We now rewrite each integral in terms of expected value. Recall that, for any measurable function $f$ and any density $p$ on $\mathcal{R}$, $\mathbb{E}_p[f] := \int_{\mathcal{R}} f(r) \, p(r) \, dr$ Therefore, for each treatment group $t \in \{0,1\}$ and cluster $k \in \{1, \ldots, K\}$, we have:

$$\int_{\mathcal{R}} f(r) \, p_{\Phi}^{t,k}(r) \, dr \;=\; \mathbb{E}_{p_{\Phi}^{t,k}}[f]$$

Using this notation, the IPM becomes:

$$\mathrm{IPM}_{\mathcal{G}}(p_{\Phi}^1, p_{\Phi}^0) = \sup_{f \in \mathcal{G}} \left| \sum_{k=1}^{K} \pi_{1,k} \, \mathbb{E}_{p_{\Phi}^{1,k}}[f] - \sum_{k=1}^{K} \pi_{0,k} \, \mathbb{E}_{p_{\Phi}^{0,k}}[f] \right|$$

4. *Isolation of local and structural contributions* At this point, the difference between treated and control mixtures combines two effects: *(i)* differences between the cluster-conditional distributions, i.e., discrepancies between treated and control units within the same cluster, $p_{\Phi}^{1,k}$ and $p_{\Phi}^{0,k}$, and *(ii)* differences between the mixture weights $\pi_{1,k}$ and $\pi_{0,k}$ (how much treated and controls end up in different clusters). To separate these contributions, we add and subtract the intermediate term $\sum_{k=1}^{K} \pi_{1,k} \, \mathbb{E}_{p_{\Phi}^{0,k}}[f]$ inside the absolute value:

$$\mathrm{IPM}_{\mathcal{G}}(p_{\Phi}^1, p_{\Phi}^0) = \sup_{f \in \mathcal{G}} \left| \sum_{k=1}^{K} \pi_{1,k} \, \mathbb{E}_{p_{\Phi}^{1,k}}[f] - \sum_{k=1}^{K} \pi_{0,k} \, \mathbb{E}_{p_{\Phi}^{0,k}}[f] \right|$$

$$= \sup_{f \in \mathcal{G}} \left| \underbrace{\sum_{k=1}^{K} \pi_{1,k} \left( \mathbb{E}_{p_{\Phi}^{1,k}}[f] - \mathbb{E}_{p_{\Phi}^{0,k}}[f] \right)}_{\text{within-cluster (local) discrepancy}} + \underbrace{\sum_{k=1}^{K} (\pi_{1,k} - \pi_{0,k}) \, \mathbb{E}_{p_{\Phi}^{0,k}}[f]}_{\text{across-cluster (structural) imbalance}} \right|.$$

5. *Triangle inequality and separation of contributions.* At this stage, the global discrepancy is expressed as the supremum of the sum of two terms: a *local* discrepancy within clusters and a *structural* imbalance across clusters. To upper bound this quantity, we apply the triangle inequality and basic properties of the supremum.

Let us define, for a fixed test function $f \in \mathcal{G}$,

$$A_f := \sum_{k=1}^{K} \pi_{1,k} \left( \mathbb{E}_{p_{\Phi}^{1,k}}[f] - \mathbb{E}_{p_{\Phi}^{0,k}}[f] \right), \qquad B_f := \sum_{k=1}^{K} (\pi_{1,k} - \pi_{0,k}) \, \mathbb{E}_{p_{\Phi}^{0,k}}[f].$$

$A_f$ captures the discrepancy between treated and control distributions *within each cluster*, while $B_f$ captures the discrepancy due to different cluster occupancies between the two groups. By the triangle inequality, for any $f \in \mathcal{G}$, $|A_f + B_f| \le |A_f| + |B_f|$, and so:

$$\mathrm{IPM}_{\mathcal{G}}(p_{\Phi}^1, p_{\Phi}^0) = \sup_{f \in \mathcal{G}} |A_f + B_f| \le \sup_{f \in \mathcal{G}} |A_f| + \sup_{f \in \mathcal{G}} |B_f|. \tag{12}$$

We now bound the two terms separately. Since the mixture weights satisfy $\pi_{1,k} \geq 0$ and $\sum_{k=1}^{K} \pi_{1,k} = 1$, we can use the subadditivity of the supremum to obtain

$$
\begin{aligned}
\sup_{f \in \mathcal{G}} |A_f| = \sup_{f \in \mathcal{G}} \left| \sum_{k=1}^{K} \pi_{1,k} \left( \mathbb{E}_{p_\Phi^{1,k}}[f] - \mathbb{E}_{p_\Phi^{0,k}}[f] \right) \right| \\
\leq \sum_{k=1}^{K} \pi_{1,k} \sup_{f \in \mathcal{G}} \left| \mathbb{E}_{p_\Phi^{1,k}}[f] - \mathbb{E}_{p_\Phi^{0,k}}[f] \right| \\
= \sum_{k=1}^{K} \pi_{1,k} \, \mathrm{IPM}_{\mathcal{G}}\big(p_\Phi^{1,k}, p_\Phi^{0,k}\big).
\end{aligned}
\tag{13}
$$

The second term is left in compact form and defines the residual *structural* discrepancy:

$$
\delta(m) := \sup_{f \in \mathcal{G}} \left| \sum_{k=1}^{K} (\pi_{1,k} - \pi_{0,k}) \, \mathbb{E}_{p_\Phi^{0,k}}[f] \right|.
$$

Combining equation 12 and equation 13, we finally obtain

$$
\mathrm{IPM}_{\mathcal{G}}(p_\Phi^1, p_\Phi^0) \leq \sum_{k=1}^{K} \pi_{1,k} \, \mathrm{IPM}_{\mathcal{G}}\big(p_\Phi^{1,k}, p_\Phi^{0,k}\big) + \delta(m)
\tag{14}
$$

### 3. Conclusion of the Proof

By combining the IPM decomposition with the generalization bound for factual and counterfactual error Shalit et al. (2017), we obtain the final formulation of Theorem 3:

$$
\mathrm{EMSE}_\Phi \leq 2 \left[ \varepsilon_{F,1} + \varepsilon_{F,0} + B_\Phi \left( \sum_k \pi_k \mathrm{IPM}_{\mathcal{G}}(p_\Phi^{1,k}, p_\Phi^{0,k}) + \delta(m) \right) \right]
\tag{15}
$$

This proves that `HERMES` optimizes the error not through indiscriminate global alignment, but through a *structured local alignment* that preserves the essential causal distinctions of the latent space. The margin does not aim to eliminate global distributional differences, but to ensure that treated and control units are aligned only where their treatment effects are comparable. This transforms the CATE estimation problem from global *extrapolation* into structured local *interpolation*.

## B   Additional Results

### B.1   Methodological Comparison of HERMES with Other Deep Learning Methods

Table 8 describes other state of the art CATE deep learning methods according to: *(i) Learning loss*, i.e., the core strategy for the loss function definition, *(ii) Dist. criterion*, i.e., criterion for the definition of treated/control distributions, *(iii) Balancing*, i.e., what is the basis for the balancing (global alignment only, neighbourhood-based alignment), *(iv) Latent space*, i.e., what notion of similarity defines "proximity", and *(v) Weakness*, i.e., the most critical weakness.

### B.2   Illustrative Example

To gain an intuitive understanding of how our method works, we provide an illustrative example. Let us consider the problem of choosing the optimal psychiatric drug, which represents a paradigmatic case where an accurate estimation of ITE is crucial. In mood disorders, many patients discontinue therapy prematurely or refuse to change molecules due to side effects or perceived low efficacy. Having a system that, based on clinical characteristics, indicates the treatment with the highest probability of benefit could improve therapeutic adherence and, consequently, health outcomes.

Table 8: Comparative analysis of CATE estimation by representation learning models

| Model | Learning loss | Dist. criterion | Balancing | Latent space | Weakness |
|---|---|---|---|---|---|
| **TARNet** (Shalit et al., 2017) | *factual loss for outcome estimation* | *factual outcome* | *no explicit alignment between treated/control* | *unstructured* | *selection bias, counterfactual extrapolation* |
| **CFRNet** (Shalit et al., 2017) | *treated/control discrepancy penalty* | *treated/control distribution discrepancy* | *covariate similarity* | *covariate similarity based-structure* | *expensive loss computationally* |
| **BCAUSS** (Tesei et al., 2023) | *self-supervised balancing score* | *treated/control weighted covariate means discrepancy* | *global covariate* | *covariate similarity-based structure* | *learned weights quality-related performance* |
| **CITE** (Li & Yao, 2022) | *softmax-based contrastive loss* | *external, static propensity score* | *global covariate* | *propensity score similarity-based structure* | *static pairs, external model* |
| **Our proposal** | *SNN margin-based contrastive loss* | *internal, dynamic ITE predictions* | *global/local ITE* | *ITE similarity-based structure* | *warm-up phase; early pairs labeling noise* |

To isolate the methodological effects, a *simulated* dataset was created as follows. We remark that, the treatment and outcome generators defined to build such a dataset, have been calibrated to, on one hand, violate the *positivity* assumption, by creating regions in the covariate space where only one of the two treatments is (almost) observable, and on the other hand, introduce *confounding* to test the robustness. Each patient is described by four key *confounders* variables, i.e., which influences both the probability of prescription and the clinical outcome: *(i) depression severity* $X_1 \sim \texttt{TruncatedNormal}(\mu = 15, \sigma = 5, a = 0, b = 30)$; *(ii) age* $X_2 \sim \texttt{TruncatedNormal}(\mu = 45, \sigma = 25, a = 18, b = 85)$; *(iii) substance abuse* $X_3 \sim \texttt{Bernoulli}(p = 0.2 + 0.015 \cdot X_1)$; *(iv) access to care* $X_4 \in \{\texttt{low}, \texttt{high}\}$ with $\Pr(X_4 = \texttt{high} \mid X_2) = \sigma(-1.5 + 0.03 \cdot X_2)$, where $\sigma(\cdot)$ is the sigmoid function.

The probability of receiving the psychiatric drug ($T = 1$) is defined as follows:

$$\Pr(T = 1 \mid \mathbf{X}) = \sigma(-1.2 + 0.30 \cdot X_1 + 0.02 \cdot X_2 - 0.8 \cdot \mathbf{1}_{\{X_3=1\}} + 1.1 \cdot \mathbf{1}_{\{X_4=\texttt{high}\}})$$

so that more serious patients with better access to care are treated more frequently, while substance abuse reduces the likelihood of psychiatric drug prescriptions.

The binary outcome $Y$ (*symptom improvement* within 6 months) is defined as follows:

$$\Pr(Y = 1 \mid T, \mathbf{X}) = \sigma\Big(-0.5 - 0.4 \cdot \frac{X_1}{30} + 0.03 \cdot X_2 - 0.8 \cdot \mathbf{1}_{\{X_3=1\}}$$
$$+ 1.0 \cdot \mathbf{1}_{\{X_4=\texttt{high}\}} + 0.6 \cdot T - 0.2 \cdot T \cdot \mathbf{1}_{\{X_3=1\}}\Big) \quad (16)$$

In this way, the drug *reduces* severity for most patients (+0.6 main effect) and the effect is attenuated in individuals with substance abuse ($-0.2$ interaction term). The dataset obtained, characterized by positivity violation and confounding, will be used to compare `HERMES` with baseline methods.

Figure 4 shows the covariates and outcome distributions in a *RCT* scenario, with 200,000 individuals of which 100,000 treated and 100,000 control. The distributions of *depression severity*, *age*, *substance abuse*, and *access to care* are nearly identical between the *Drug* and *No Drug* groups. Randomization breaks any association between covariates and treatment assignment, allowing the difference in outcome (*symptom improvement*) to be interpreted as *causal effect* of drug.

Figure 5 shows the distributions in an *Observational* scenario (OBS), with 200,000 individuals of which 142,000 treated and 58,000 control. As we can see, patients receiving the drug tend to have more severe depression and higher average age; conversely, substance abuse and limited access to care are more frequent among the untreated. This pattern, known as *confounding by indication*, produces the paradox that the drug appears to worsen symptoms. In reality, the negative effect is due to the fact that the treatment is assigned to the most compromised patients from the beginning.

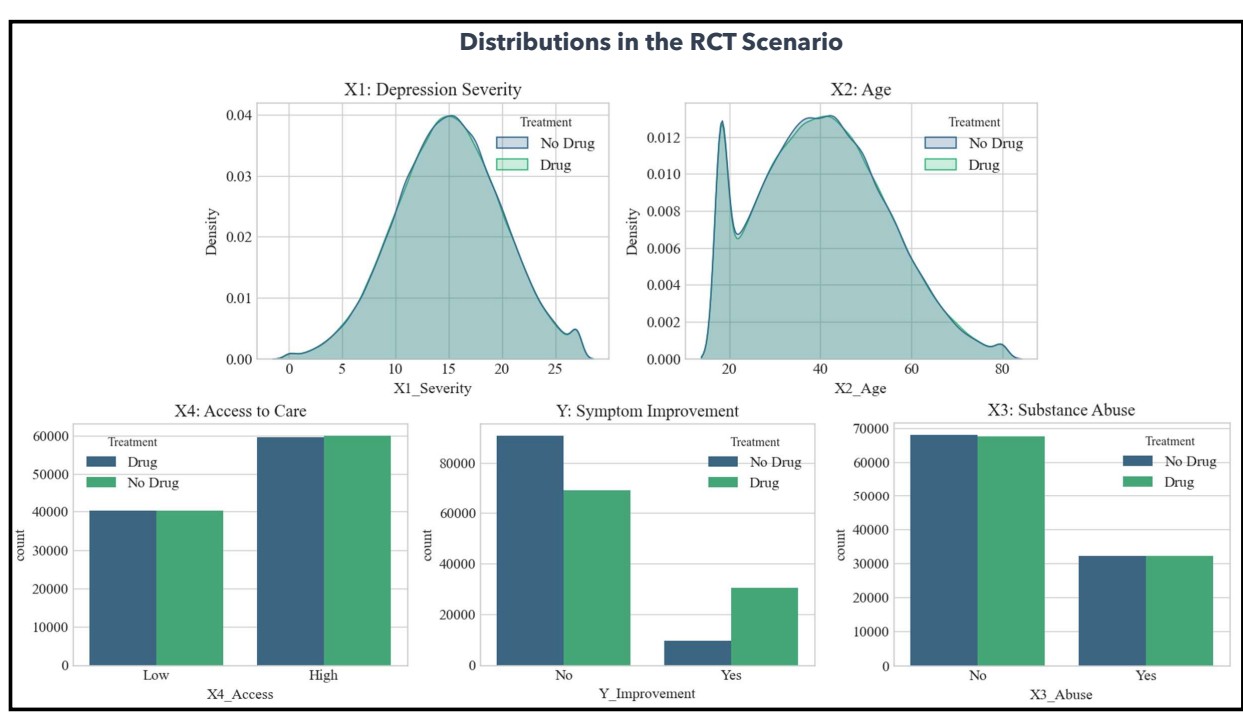

Figure 4: *RCT scenario*: comparison of covariates and outcome distributions.

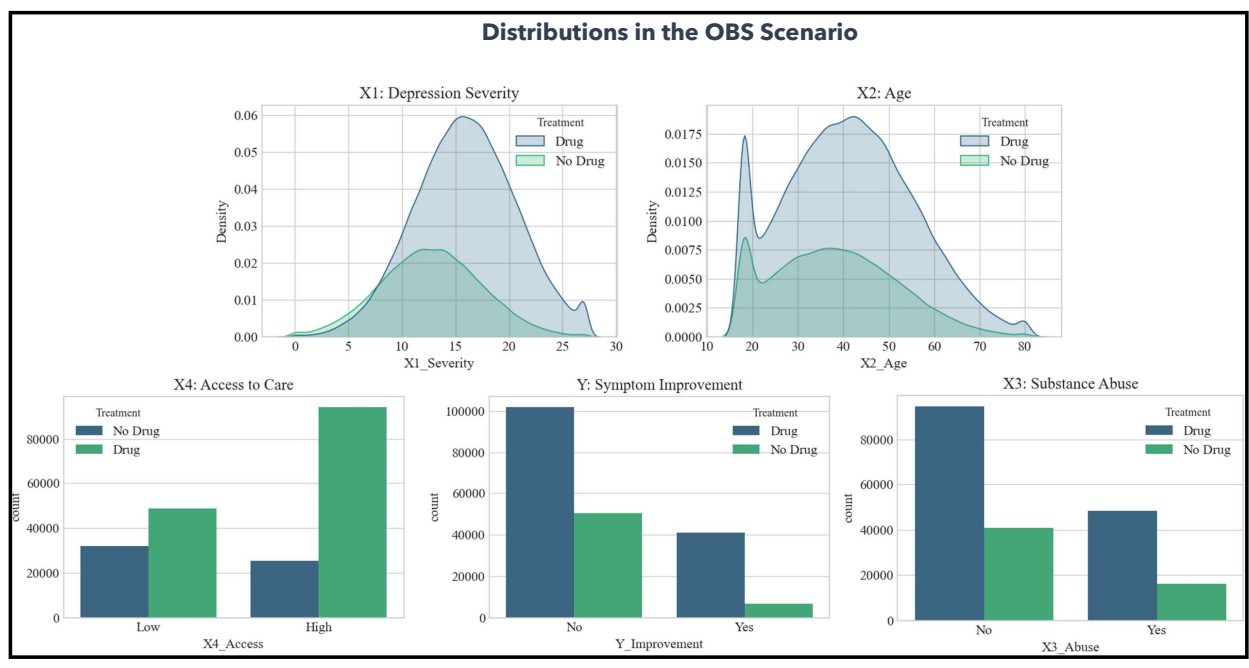

Figure 5: *OBS scenario*: comparison of covariates and outcome distributions.

In the OBS scenario, propensity-score methods are often used to adjust for measured confounding and improve treated–control comparability. However, the function used to estimate such a probability affects the ability of the model to balance the distributions. Figure 6 shows two hypothetical functions analyzed in Tesei et al. (2023):

1. *BCE.* Binary cross-entropy "step" function that learns a threshold ($\approx 15$ in Figure 6) on the *depression severity* covariate, above which $P(T{=}1 \mid X) = 1$, by correlating covariates to treatment assignment rather than the treatment effect; this produces extreme propensity scores, and highly dissimilar latent representations for treated versus control individual.

2. *BCAUSS.* Softer loss balancing function on the weighted representation of covariates: the probability of treatment varies continuously with the depression severity covariate, avoiding threshold-based decisions; the propensity scores are more overlapped on the covariates, indicating greater covariate balance in the latent space and more favorable conditions for estimating the causal effect (less residual confounding); by reducing distributional discrepancies between treated and control representations, i.e., enforcing $p(X \mid T{=}1) \approx p(X \mid T{=}0)$ for the learned representations, `BCAUSS` makes treatment assignment nearly independent of pre-treatment covariates conditional on $X$ (approximately $T \perp (Y(0), Y(1)) \mid X$).

Figure 6 shows propensity function and propensity score distributions under the two functions described above. For both such functions, the "left column" highlights the relationship between the *depression severity* covariate and the estimated treatment probability $P(T{=}1 \mid X)$, while the "right column" shows the histograms of the resulting propensity scores for the treated and control groups. As we can see on the "top", in *BCE*, the model learns a step function with a threshold at $\sim 15$, assigning treatment almost deterministically above the threshold and almost never below it. This yields extreme propensity scores (near 0 or 1) and little overlap between groups. On the "bottom", `BCAUSS` learns a balancing function over a weighted representation of the covariates, so treatment probability varies continuously with severity. The resulting propensity scores show substantially more overlap, indicating better covariate balance between treated and control individuals.

Our network extends the idea of balancing by introducing a *contrastive loss* that operates directly on the ITE. Each anchor sample is associated with: a *positive point* with a similar ITE; a *negative point* with a different ITE. As shown in Figure 7, the training encourages the network to map anchors and positives within a sphere of radius $m$, while keeping negatives outside this margin. By combining balancing and contrastive learning, our architecture allows for robust estimation of individual treatment effects.

### B.3 Stress Tests: Noisy Early Estimates, Lower Overlap, and Stronger Confounding

To further assess the robustness of `HERMES` beyond the standard benchmarks, we conducted a set of empirical stress tests designed to target three challenging aspects of observational CATE estimation: *(i)* noisy early pseudo-ITE estimates, *(ii)* reduced overlap, and *(iii)* progressively stronger confounding. The goal is to evaluate whether the self-supervised feedback loop remains stable and whether the proposed contrastive mechanism retains its advantage precisely in the regimes where counterfactual estimation is most exposed to extrapolation.

All tests were conducted on the same simulated scenario introduced in Section B.2. Starting from that base data-generating process, we change two parameters in order to create progressively more difficult observational regimes: an *overlap parameter*, which makes treatment assignment more or less deterministic, and a *confounding parameter*, which scales the influence of prognostic covariates on both treatment and outcome. This creates a $3 \times 3$ grid of conditions combining overlap levels *high / medium / low* with confounding levels *low / medium / high*, while preserving the same heterogeneous treatment-effect structure described above.

For each combination of overlap and confounding, we compare three models:

1. `FactualOnly`, a baseline trained only with the factual regression loss;

2. `HERMES`, trained with the standard contrastive procedure after a 20-epoch warm-up;

3. `HERMES + Noise`, in which Gaussian noise ($\sigma = 2.0$) is injected into the pseudo-ITE estimates immediately before pair mining, after the warm-up phase.

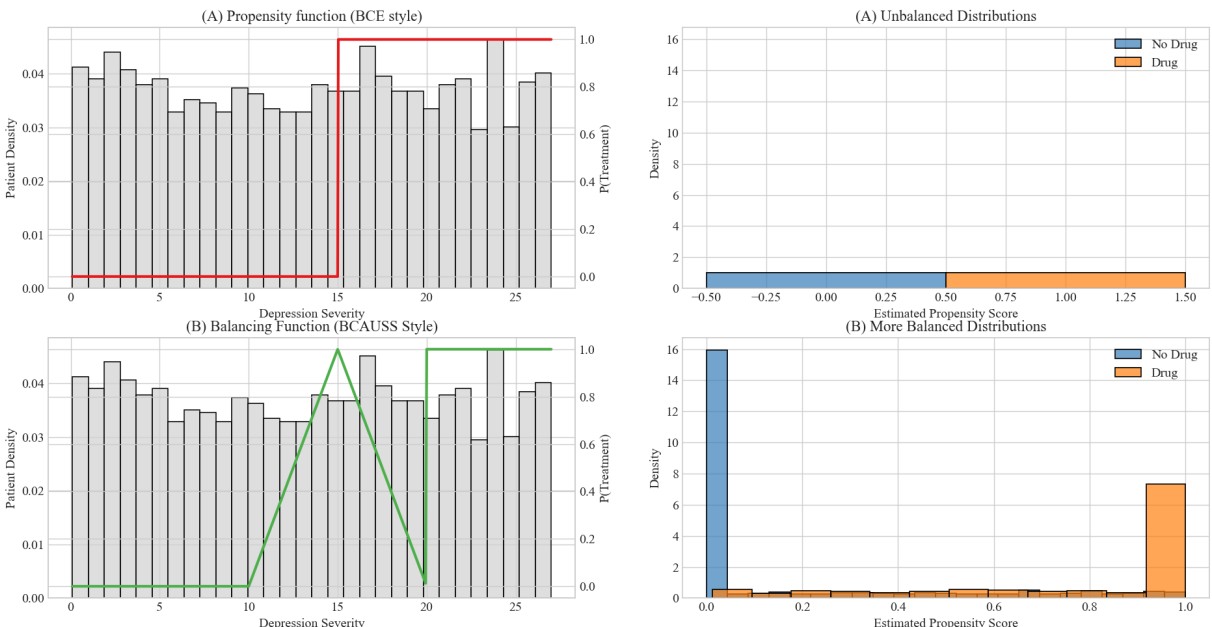

Figure 6: Propensity function and propensity–score distributions under two training objectives. *Left column:* relationship between the covariate *depression severity* and the estimated treatment probability $P(T=1 \mid X)$. *Right column:* histograms of the resulting propensity scores for the treated (*Drug*) and control (*No Drug*) groups. *Top row (BCE).* The model trained with Binary Cross-Entropy learns a near step function with a threshold at $\sim 15$, assigning treatment almost deterministically above the threshold and almost never below it. This yields extreme propensity scores (near 0 or 1) and little overlap between groups. *Bottom row (BCAUSS).* The BCAUSS objective learns a smoother balancing function over a weighted representation of the covariates, so treatment probability varies continuously with severity. The resulting propensity scores show substantially more overlap, indicating better covariate balance between treated and control units.

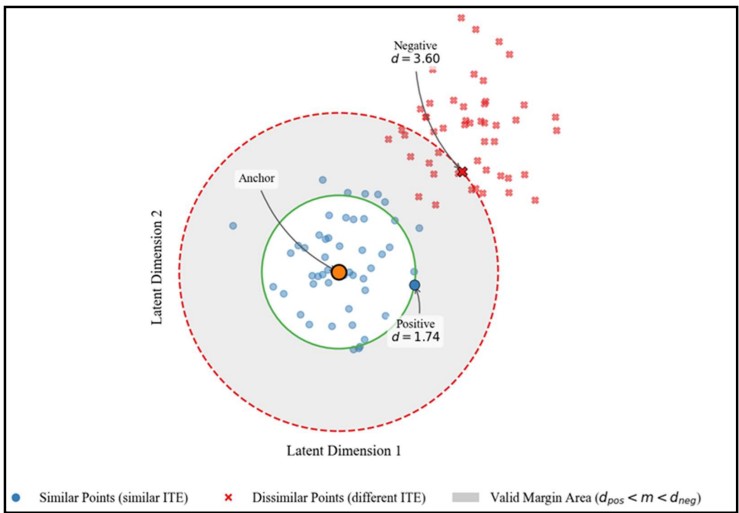

Figure 7: Impact of the *margin m* in the HERMES's contrastive loss. Each **anchor** (solid circle) is paired with a **positive** with a similar ITE (blue circle) and a **negative** with a dissimilar ITE (red cross). The loss forces positive pairs to be *inside* a sphere of radius $m$ and negative pairs *outside*.

Each configuration is repeated across multiple random seeds. We report the out-of-sample ITE RMSE both on the full test set and on the subset of *low-support* units, defined as those with extreme propensity values.

This allows us to directly evaluate not only global estimation quality, but also the behavior of the models in the most difficult regions of the covariate space.

Table 9 reports the evolution of the out-of-sample ITE RMSE during training with and without artificial noise injection. As expected, once the contrastive phase starts, heavy corruption of the pseudo-ITE estimates produces an immediate degradation in performance. However, the model does not collapse into a terminal negative feedback loop. Because the factual loss remains active throughout training, the outcome heads provide a persistent anchor that gradually counteracts the noisy geometric constraints. By the end of training, the stress-test model remains highly competitive with the uncorrupted baseline, showing that the self-supervised pairing mechanism is robust to early perturbations.

Table 9: Evolution of ITE RMSE during training with and without artificial noise injection in the contrastive pairing phase.

| Epoch | Phase | Baseline (Noise 0.0) | Stress Test (Noise 2.0) |
|---|---|---|---|
| 0 | Warm-up | 1.0812 | 1.075 |
| 20 | Contrastive (Start) | 0.2345 | 0.2236 |
| 40 | Contrastive | 0.1686 | 0.2809 |
| 60 | Contrastive | 0.2838 | 0.2376 |
| 80 | Contrastive (End) | 0.1900 | 0.1828 |

Table 10 summarizes the systematic robustness experiment. Two patterns are particularly clear. First, in easy regimes, where overlap is high and confounding is weak, all models are competitive and the advantage of contrastive regularization is limited. This is expected, since under favorable conditions the estimation problem already resembles interpolation.

Second, as overlap deteriorates and confounding becomes stronger, the gap between the methods widens substantially. In the medium-overlap/high-confounding regime, the average RMSE drops from 0.430 for `FactualOnly` to 0.143 for `HERMES`. In the low-overlap/medium-confounding regime, the reduction is even larger, from 0.426 to 0.137. In the hardest regime (low overlap, high confounding), `HERMES` reduces the average RMSE from 0.713 to 0.387. Importantly, the same pattern is observed on the low-support subset, where counterfactual estimation is most fragile.

Finally, the noisy variant remains close to standard `HERMES` across conditions, with only moderate degradation in the hardest settings. This suggests that the warm-up phase and the persistent factual anchor successfully stabilize the self-supervised feedback loop even when the early pseudo-ITE estimates are severely corrupted.

Table 10: Systematic robustness under different overlap and confounding regimes. Values are reported as mean RMSE over repeated runs. Lower is better.

| Overlap | Confounding | FactualOnly | HERMES | HERMES + Noise |
|---|---|---|---|---|
| Medium | High | 0.430 | **0.143** | 0.156 |
| Low | Medium | 0.426 | 0.137 | **0.121** |
| Low | High | 0.713 | **0.387** | 0.440 |
| *Average over all regimes* | | 0.244 | **0.14** | **0.140** |

Overall, these experiments support two conclusions. First, the gain of `HERMES` is not limited to standard benchmark settings, but becomes most visible when overlap is reduced and confounding is strong. Second, the self-supervised contrastive mechanism remains stable even under heavy perturbations of the early pseudo-ITE estimates. These results provide additional evidence that the proposed strategy is particularly beneficial in the regimes where counterfactual estimation is most exposed to unreliable extrapolation.

# C   Additional Benchmark Analysis on ACIC 2016

To complement the main benchmark evaluation on IHDP and Jobs, we further assessed the proposed model on the ACIC 2016 benchmark. ACIC 2016 was introduced in the context of the Atlantic Causal Inference Conference Data Analysis Challenge and provides a semi-synthetic benchmark built from real covariates and simulated treatment-response mechanisms (Dorie et al., 2019). The benchmark is based on covariates derived from the Collaborative Perinatal Project, while treatment assignment and potential outcomes are simulated so as to preserve realistic covariate structure while still providing access to ground-truth causal effects.

ACIC 2016 is used as a robustness benchmark for treatment-effect estimation because it spans multiple data-generating regimes with varying levels of confounding, overlap, response-surface complexity, and treatment-effect heterogeneity. In its commonly used version, the benchmark comprises 77 distinct data-generating settings, each with repeated replications. After standard preprocessing and one-hot encoding of categorical variables, the resulting input dimensionality is commonly reported as 82 covariates (Curth & Van der Schaar, 2021).

We considered the collection of 77 benchmark settings and evaluated the proposed model across all settings. Performance was summarized using the RMSE and ATE Error, defined as in Section 4.2. We report this metric as a global summary over the benchmark settings under our evaluation pipeline.

## C.1   Results

Across the 77 settings of ACIC 2016, the proposed model achieved a RMSE of 1.951 and ATE error af 0.20. This result provides additional evidence that the proposed representation remains effective beyond IHDP and Jobs, including on a broader benchmark characterized by more heterogeneous and structurally complex observational regimes.

Direct numerical comparison with previously published ACIC 2016 results should be interpreted with caution. In the ACIC literature, reported performance often depends on the evaluation target and experimental protocol, including whether the target is an individual-level treatment effect or an aggregate estimand, as well as on preprocessing choices, train/test splitting, covariate transformations, and aggregation across simulation settings (Dorie et al., 2019). For this reason, we view the ACIC analysis primarily as a complementary robustness evaluation under our own experimental pipeline rather than as a definitive comparison against prior work.

Overall, the ACIC 2016 evaluation should be interpreted as a complementary large-scale robustness check rather than as the primary benchmark comparison of the paper. Together with the main IHDP and Jobs results, it supports the conclusion that organizing the latent space according to treatment-effect similarity can be beneficial not only on standard benchmarks, but also in broader semi-synthetic settings characterized by substantial heterogeneity and nonlinear structure.

