# OpenReview forum: "HERMES: Heterogeneous Effects Representation with Matched Embeddings using Siamese Networks"
_TMLR — Accepted by TMLR_

### Review · Reviewer_RpgT · 2026-03-14

**Summary Of Contributions:**

This paper proposes a Siamese neural network architecture, named HERMES, for estimating the Conditional Average Treatment Effect (CATE) from observational data. The method leverages representation learning together with self-supervision and contrastive learning, and introduces a novel contrastive loss that organizes the latent space based on the similarity between estimated individual treatment effects (ITEs), rather than relying solely on covariates. This approach aims to learn globally and locally balanced representations of covariate information, leading to more robust CATE estimation. Experimental results show that the proposed method reduces the MSE by 14–15\% compared with baseline methods on the IHDP and JOBS benchmarks.

**Additional Comments:**

1. The abstract provides a general overview of the proposed method and experimental results. However, it could be improved in terms of clarity and accessibility, particularly for readers who are not deeply familiar with the specific subfield of causal representation learning.

**Audience:**

Yes

**Audience Explanation:**

YES.

The paper addresses an important problem in causal machine learning, namely the estimation of conditional average treatment effects (CATE) from observational data. Representation learning methods for causal inference (e.g., TARNet, CFRNet, and DragonNet) are widely studied in the machine learning community. The proposed idea of structuring latent representations using contrastive learning based on estimated treatment-effect similarity represents an interesting direction that combines ideas from self-supervised learning and causal inference. Researchers working on causal representation learning and treatment effect estimation would likely find the approach relevant. The integration of contrastive learning into causal representation learning could inspire further research in this area.

**Broader Impact Concerns:**

The proposed method is intended for applications such as policy evaluation, healthcare decision making, and personalized treatment, where estimating heterogeneous treatment effects is important. While the method itself does not introduce direct societal risks, models used for treatment-effect estimation may influence important decisions (e.g., medical treatments or policy allocation).

**Claims And Evidence:**

Yes

**Claims Explanation:**

YES (partially).

The paper proposes HERMES, a Siamese neural network that utilizes a contrastive loss to learn latent representations structured by estimated individual treatment effects (ITE). The submission presents strong overall empirical results (e.g., lower RMSE on IHDP and reduced Policy Risk on JOBS), which generally support its claims of improved performance.

Specifically, the paper lacks a formal ablation study. Because the base architecture borrows heavily from existing frameworks (e.g., BCAUSS), it is critical to isolate the impact of the newly introduced components. Currently, the paper lacks a formal ablation study, making it difficult to determine whether the performance gains stem from the novel margin-based contrastive loss and dynamic pair mining strategy, or merely from the baseline capacity and hyperparameter tuning. While the claims are generally supported, the experimental validation must be expanded.

**Requested Changes:**

1. It would be helpful if the authors could briefly outline the structure of the paper at the end of the introduction.

2. The relationship between Theorem 2 and Theorem 3 is not entirely clear. While Theorem 3 introduces a cluster-wise IPM decomposition with an additional term $\delta(m)$, it is not obvious from the formulation whether this bound is tighter than the one in Theorem 2 or under what conditions it becomes advantageous. Providing additional theoretical discussion or intuition clarifying the relationship between the two bounds would strengthen the theoretical section.

3. The paper would benefit from copy editing. There are typos, for example in Section 4.1 ``metrics for checking the causal assumptions introduced in 2.''.

4. While HERMES demonstrates strong empirical results, it lacks an ablation study to isolate the impact of its novel components. Given that the foundational BCAUSS model provided a detailed ablation table (Table 2) to validate its individual loss terms, HERMES should follow suit. Please provide a similar ablation table that explicitly quantifies the individual contributions of the margin-based contrastive loss ($\mathcal{L}_{ctr}$) and the dynamic pair mining strategy.

---

> ### Author Response · Authors · 2026-04-26
> **Response to reviewer RpgT**
>
> We thank the reviewer for the careful and constructive feedback. We revised the paper by adding a formal ablation study and clarifying the theoretical discussion.
>
> 1) We agree that a formal ablation study was necessary to isolate the contribution of the components newly introduced in HERMES. We added a dedicated ablation study. In particular, we evaluate variants that remove or modify the contrastive objective and the dynamic pairing mechanism. We also compare the proposed dynamic pseudo-ITE pairing against random pairing, covariate-based pairing, and static pseudo-labeling. These comparisons allow us to test whether the improvement comes from the proposed effect-aware pairing rule, rather than merely from adding a contrastive regularizer. The results show that the full HERMES configuration achieves the best overall performance among the considered variants. In particular, pairing units according to evolving pseudo-ITE similarity leads to more effective representation structuring than random or purely covariate-based criteria, while also outperforming static pseudo-labeling, which cannot adapt as the model improves during training. This supports the claim that the observed gains are tied to the proposed dynamic, effect-aware pair mining strategy and margin-based contrastive loss, rather than only to baseline capacity or tuning.
> Position: Section 5.3 “Ablation Study” (pp. 19–21).
>
> 2) We agree that a brief roadmap at the end of the Introduction improves readability.
> Position: Section 1 “Introduction” (p. 3).
>
> 3) We agree that the relationship between Theorem 2 and Theorem 3 required further clarification. In the revised paper, we added discussion after Theorem 3 to explain their respective roles and the intuition behind the cluster-wise IPM decomposition. Theorem 3 should not be interpreted as uniformly tighter than Theorem 2 in all possible settings. Rather, its contribution is conceptual and structural. Theorem 2 provides a global discrepancy-based perspective, consistent with standard representation-learning bounds for counterfactual estimation. Theorem 3 instead decomposes the global discrepancy into cluster-level discrepancies together with an additional term accounting for cross-cluster mismatch. This decomposition becomes informative when the learned representation induces locally coherent groups. In such cases, the bound helps explain why organizing the latent space around effect-relevant neighborhoods may be beneficial: local discrepancies can be controlled within meaningful regions of the representation space, while the additional term captures the cost of mismatch across regions. We therefore present Theorem 3 not as a universal tightening,
> but as a way to formalize the role of local structure in the proposed method.
> Position: Section 3.2 “Contrastive Regularization and IPM Reduction” (pp. 12–13).
>
> 4) We revised the manuscript to correct typographical errors and improve clarity and readability throughout the paper. This includes the sentence noted by the reviewer in Section 4.1, which has been corrected to avoid ambiguity.
> Position: Throughout the manuscript, including Section 4.1.
>
> 5) We agree that applications such as policy evaluation, healthcare decision making, and personalized treatment require an explicit discussion of potential risks. Although HERMES does not introduce direct societal risks by itself, treatment-effect models can influence consequential decisions if used for medical treatment assignment, resource allocation, or policy design. We therefore added a broader-impact discussion at the end of the paper. The revised discussion emphasizes that CATE estimates should not be used as standalone decision rules in high-stakes domains without careful validation, uncertainty assessment, domain expertise, and checks for dataset bias and subgroup performance. We also clarify that the method is intended as an estimation tool, not as an autonomous decision-making system, and that deployment in sensitive settings requires appropriate human oversight and domain-specific evaluation.
> Position: Section “Broader Impact” (p. 25).
>
> 6) We agree that the abstract could be clearer and more accessible, particularly for readers who are not deeply familiar with causal representation learning. We therefore revised the abstract to present the motivation, the methodological contribution, and the empirical findings more transparently. In particular, we clarified that HERMES uses dynamically generated pseudo-ITE pair labels to structure the latent space according to treatment-effect similarity. At the same time, we retained the technical terminology needed to describe the contribution precisely, since the paper targets CATE estimation and causal representation learning.
> Position: Abstract.

---

### Review · Reviewer_jsfq · 2026-03-22

**Summary Of Contributions:**

The paper proposes HERMES, which essentially tries to move away from learning representations based purely on covariate similarity and instead structure the latent space around treatment-effect similarity. It uses a network with a contrastive loss forming pairs based on the model’s estimated ITE, so the representation is gradually shaped to bring together units with similar predicted effects and separate those with different ones.

The idea is that this creates more meaningful local neighborhoods for counterfactual comparison, turning the problem from global balancing into something closer to local interpolation. Empirically it shows solid improvements on IHDP and JOBS, and the latent space diagnostics are consistent with the intended behavior.

**Audience:**

Yes

**Audience Explanation:**

Yes, since the method is a relatively simple add-on, it can be incorporated into many existing CATE/ITE frameworks with minimal changes. The approach is intuitive and aligns well with the goal of learning effect-relevant representations, which makes it appealing to practitioners and researchers working on causal representation learning.

**Claims And Evidence:**

Yes

**Claims Explanation:**

The empirical results on standard benchmarks like IHDP and JOBS solidly show improvements over baselines, which supports the main claims. The latent space diagnostics also match the intuition of the method.

**Requested Changes:**

Some of the paper’s stronger claims, particularly around reduced extrapolation and improved causal validity, are supported mainly by intuition and indirect evidence rather than direct theoretical or empirical validation. While the appendix provides additional motivation, it does not establish when the proposed mechanism is reliable or how it behaves under challenging conditions.

A key limitation is the lack of ablation studies isolating the contribution of the contrastive pairing strategy. The paper does not compare against alternatives such as random pairing, covariate-based similarity, or static versus dynamic pseudo-labeling, making it difficult to attribute performance gains specifically to the proposed method.

In addition, the evaluation is limited to standard benchmarks. Given that the approach relies on self-estimated ITE to construct training signals, it would be important to assess robustness under settings with lower overlap, stronger confounding, or noisy early estimates.

The paper also does not analyze the stability of this feedback process or discuss potential failure modes, which would be important for establishing confidence in the method.

---

> ### Author Response · Authors · 2026-04-26
> **Response to reviewer jsfq**
>
> We thank the reviewer for the careful and constructive feedback. We revised the paper to provide more empirical support
> for the claims, to strengthen the ablation analysis, and to clarify the stability and failure modes of the dynamic pairing mechanism.
>
> 1) We agree that the previous version relied too heavily on intuition when discussing reduced extrapolation and improved causal validity. We added a dedicated analysis of local support in the learned latent space. We now report three diagnostics: (i) the nearest opposite-treatment distance measures how far each unit is from its closest latent neighbor receiving the opposite treatment, (ii) the support-gap ratio quantifies the relative increase in distance when moving from same-treatment to opposite-treatment neighbors, (iii) the hard-gap ratio measures the fraction of units for which the closest opposite-treatment neighbor remains substantially farther away than the closest same-treatment neighbor. These metrics provide an assessment of local extrapolation beyond standard global balance metrics. The results show that HERMES improves local support compared with the considered alternatives and produces latent neighborhoods in which opposite-treatment comparisons are more accessible.
> Position: Sec. 4.5 “Local Support and Extrapolation Diagnostics” (pp. 17–18); Sec. 5.4 “Latent Space Analysis” (p. 23).
>
> 2) We agree that an ablation study was necessary to isolate the role of the proposed pairing strategy. We therefore added a dedicated ablation analysis comparing dynamic pseudo-ITE pairing against random pairing, covariate-based similarity, and static pseudo-labeling. This separates the effect of the proposed pairing rule from the mere presence of a contrastive objective.
> The results show that the dynamic effect-aware strategy achieves the best overall performance among the considered variants. This supports the interpretation that the gain is not simply due to additional regularization, model capacity, or hyperparameter tuning, but to the proposed organization of the latent space according to evolving treatment-effect similarity. The ablation also clarifies the role of the dynamic component: since pseudo-ITE estimates improve during training, static pairing may preserve early errors and impose an outdated geometry, whereas dynamic refreshing allows the contrastive signal to evolve with the model.
> Position: Sec. 5.2 “Ablation Study” (pp. 19–21).
>
> 3) We agree that robustness should be assessed beyond standard benchmark conditions, especially because HERMES uses self-estimated ITEs to construct part of its training signal. We therefore added tests targeting lower overlap, stronger selection bias, and noisy early pseudo-ITE estimates. These experiments evaluate whether the method remains stable when the self-supervised pairing signal is less reliable. The tests provide two main findings. First, the advantage of HERMES becomes pronounced in
> harder regimes with weak overlap and stronger selection bias, suggesting that the contrastive mechanism is particularly useful when counterfactual estimation is more difficult and local support is limited. Second, when early pseudo-ITE estimates are artificially corrupted, the method does not collapse into a degenerate feedback loop. Although performance becomes more challeng-
> ing under corrupted signals, training remains stable because the contrastive objective is introduced only after an initial warm-up and because factual supervision remains active throughout optimization. Thus, the added tests provide evidence that the mechanism is reasonably robust when the pseudo-ITE signal is degraded.
> Position: Sec. 3.3 “Implementation” (pp. 14–15); Appendix B.3 “Stress Tests: Noisy Early Estimates, Lower Overlap, and Stronger Confounding” (pp. 33–35).
>
> 4) We agree that the stability of the feedback process and its failure modes needed clearer discussion. We now explicitly describe the main potential failure mode: if early pseudo-ITE estimates are too noisy to contain even weak local ranking information, the induced pair assignments may become unreliable and distort the latent geometry. In such cases, the contrastive objective may
> amplify noise rather than useful treatment-effect structure. We clarify that HERMES does not require accurate pseudo-ITE estimates from the beginning of training. Instead, it relies on a weaker condition: after factual warm-up, the model should provide enough relative information to identify some pairs that are more likely to have similar treatment effects than random alternatives. We also expanded the description of the stabilizing mechanisms used to reduce feedback instability, including factual warm-up, periodic rather than continuous label refreshing, threshold clamping to avoid degenerate pair assignments, and an
> always-active factual loss anchoring the representation to observed outcomes. Finally, we connect
> this discussion to the previous test.
> Position: Sec. 3.3 “Implementation” (pp. 14–15).

---

### Review · Reviewer_mjHB · 2026-04-12

**Summary Of Contributions:**

This paper discusses existing representation learning methods for ITE estimation with observational data and provides an additional perspective and framework. The contributions can be summarized as:
1. The authors discuss existing approaches that leverage the idea  of balancing covariate representations to adjust for the covariate shift between treated and control group coming from confounding/selection bias existing in observational data, which can improve estimation error for ITE estimation. Here, the paper gives an in general interesting intuition on that balancing only in covariate similarity might not be optimal, but that representation learning approaches might benefit more from balancing with respect to similarity in ITE between observations between the groups.
2. The paper proposes HERMES, a neural framework that implements this idea. Here, the authors combine existing ideas from representation learning and CATE estimation similar to the correctly referenced BCAUSS and CFRNet papers and extends them with a self-supervised contrastive objective leveraging Twin networks, similiar to the existing CITE method.
3. The paper provides some theory for their method. This this is an extension of the generalization error of ITE estimation with balancing provided by previous work to balancing for existing clusters. However, these results assume existing clusters and neglect the estimation error for determining such clusters.
4. The authors provide some experimental results on two standard semi-synthetic benchmarks showing good empirical performance compared to baselines. However, these could also be extended e.g. with the ACIC datasets.

**Audience:**

Yes

**Audience Explanation:**

In general, the approach might be interesting since balancing is still often done using only simple IPM regularization between treated and control covariate distributions. However, I am not sure whether there is enough difference to the existing papers around the prognostic scores above. Thus, the other contribution around the neural framework might be interesting, however, I think here the usefulness should be a bit more theoretically grounded or the intuition should be clearer here. Even though the empirical performance is strong, I think this might not be sufficient for being of enough interest to people in the causal inference field since this is usually used more for validation of interesting ideas than for proper benchmarking.

**Broader Impact Concerns:**

No concerns here

**Claims And Evidence:**

No

**Claims Explanation:**

I think the current version has 2 major open points that are not clear yet.
1. The general main idea, i.e. using similarity in ITE for balancing (rather than similarity in marginal covaraite distributions or propensity score) is presented as a novel idea. However, this is also the direct motivation in the literature around the “prognostic score” (Hansen 2008, https://www.jstor.org/stable/20441477) and has also been recently discussed in representation learning approaches and other directions e.g., in (https://arxiv.org/pdf/2104.05762, https://academic.oup.com/biomet/article-abstract/104/3/583/3836906, https://www.tandfonline.com/doi/full/10.1080/07350015.2019.1609974, https://arxiv.org/pdf/2604.00811, and more). Thus, the relation to this literature stream needs to be discussed, and such methods also serve as potential baselines. Overall, the contribution and novelty of this part of the paper needs to be reevaluated as a consequence.
2. The theory proposes why clustering with respect to ITE can be helpful if there is access to useful clusters. However, it is unclear how the estimation error for such clusters affects the performance. Since the method uses the ITE predicitons of the model for building pairs, intuitively I would think that this means either that the ITE predictions are already good to allow for good clustering (--> if the predictions are already good, why is the contrastive part necessary as it induces additional complexity) or the ITE estimates are not good in the first stage (--> then using these predictions for building pairs would lead to unreliable clusters, also not helping estimation). Thus, overall the intuition when and why the method can help without access to useful ground truth ITE pairs/clusters is not clear to me yet.

Further, some parts in method and problem setting description are unclear.
-  E.g., the use of confounding bias when actually the authors refer more to a finite sample estimation bias, under ignorability there is no biased due to unobserved confounding, only due to covariate shift.
- Also, in the introduction the authors state “To build such pairs, no external labels are needed: covariates X and treatment T suffice, using the ITE values computed from the model’s own predictions. The learning signal comes entirely from the data, without leveraging the observed outcome Y .” This is unclear, since the observed labels Y are needed to train the model to generate ITE predicitons in the first place, what is meant here?
- the contribution over CITE are not that clear to me, it sounds like the only difference is that HERMES does not require an additional pretrained propensity model, which does not seem to make a large difference.

**Requested Changes:**

- The major two points stated around novelty of the general idea and relation to the prognostic score literature, and improved intuition/theory regarding the framework need to be fully addressed and clarified for potential acceptance.

Further changes that would clearly improve the paper
-  I think at some point the paper is still ambiguous and does not always follow common terms in CATE estimation. E.g. in Section 2.1, they state “second, it [the representation] must remove the confounding bias.” Usually confounding bias refers to the unreducible bias due to unobserved confounding. I think here the authors mean more like the “finite-sample selection bias”, which comes from the covariate shift in the observed confounders. I think this should be aligned again properly with existing literature in multiple parts in this paper.
- In the end of section 1, BCAUSS is mentioned without being explained before.
- additional experiments e.g. with ACIC datasets, and baselines from prognostic score literature

---

> ### Author Response · Authors · 2026-04-26
> **Response to reviewer mjHB**
>
> We thank the reviewer for the careful and constructive feedback. In the following, we will answer the key questions in detail.
>
> 1) In light of the literature suggested as baseline, we agree that our original intuition should be refined and more carefully positioned. HERMES does not claim novelty for the broad idea that outcome-relevant representations can support causal adjustment. Its contribution is a dynamic self-supervised contrastive mechanism that structures the latent space by ITE similarity.
> Prognostic scores summarize outcome-relevant information under one treatment condition, e.g. expected control outcome, but similar prognosis need not imply similar treatment response. HERMES instead defines pair labels from $\hat\tau(x)=h_1(\Phi(x))-h_0(\Phi(x))$, encouraging neighborhoods to reflect causal-response similarity rather than only baseline-risk or propensity similarity.
> We revised the paper accordingly.
> Position: Sec. 2.1 (pp. 4–5); Sec. 3.1 (p. 9).
>
> 2) HERMES does not assume ground-truth ITE clusters or a fixed clustering stage. Pair labels are built dynamically from evolving pseudo-ITE estimates and periodically refreshed. Thus, after factual warm-up, pseudo-ITEs only need enough local ranking information for some pairs to be more likely effect-similar than random alternatives. The contrastive term amplifies this weak
> signal by moving effect-similar units closer and effect-dissimilar units apart. If early pseudo-ITEs were entirely uninformative, pair labels would be noisy and the contrastive term would not help. We now discuss this failure mode and the stabilizers used in HERMES: factual warm-up, periodic refresh, and factual loss.
> Position: Sec. 3 (p. 9); Sec. 3.3 (pp. 14–15).
>
> 3) We agree that, under ignorability, the issue is not irreducible bias from unobserved confounding, but finite-sample selection bias due to covariate shift between treated and control units. We revised the terminology and now refer to balancing/deconfounding properties rather than to “removing confounding bias.”
> Position: Sec. 2.1 (p. 3).
>
> 4) The statement in the introduction referred to the pair labels used by the contrastive component of HERMES, not to the observed outcomes Y . The observed outcomes are used in the supervised factual loss to learn potential-outcome predictions, from which the model obtains pseudo-ITE estimates. In contrast, the pairwise labels required by the contrastive objective are not externally
> provided. During training, pairs are formed by comparing the model’s current pseudo-ITE predictions: units with similar predicted treatment effects are treated as positive pairs, whereas units with dissimilar predicted treatment effects are treated as negative pairs. Thus, for a given pair of units, the contrastive label is generated internally from the model’s current estimate of
> ITE similarity, rather than from an external annotation. These pair labels are not available in ground-truth form at the beginning of training. Instead, they are constructed dynamically from the model’s evolving predictions and are progressively refined as the potential-outcome predictors improve.
> Position: Sec. 1 (pp. 2–3).
>
> 5) The contribution over CITE is not merely avoiding a pretrained propensity model. The main distinction is the contrastive supervision signal. CITE forms pairs using propensity scores, inducing a geometry mainly oriented toward overlap and treatment-assignment balance. HERMES forms pairs using estimated ITE similarity, so latent proximity reflects treatment-response
> similarity. This matters because good overlap alone does not ensure nearby units have similar causal effects. HERMES is also modular: whenever a CATE model provides treatment-specific predictions, the same dynamic effect-based pairing strategy can structure its representation. We revised the discussion accordingly.
> Position: Sec. 2.3 (pp. 7–8).
>
> 6) We aligned the terminology with standard CATE usage throughout the manuscript. Where the previous text referred to “confounding bias,” we now distinguish between ignorability and finite-sample bias from covariate shift in observed confounders, using terms such as balancing, overlap, and selection bias.
> Position: Sec. 2.1 (p. 3).
>
> 7) We revised the introduction to briefly introduce BCAUSS before mentioning it at the end of Section 1, so that its role is clearer.
> Position: Sec. 1 (p. 2).
>
> 8) We expanded the empirical evaluation with ACIC 2016 experiments and broadened the related work discussion on prognostic-score methods. On ACIC 2016, HERMES remained competitive under our evaluation pipeline, achieving a global treatment-effect RMSE of 1.951 across the 77 benchmark settings. We report this cautiously, since ACIC comparisons are sensitive to protocol, simulation selection, and aggregation, but it provides additional evidence beyond IHDP and Jobs.
> Position: Appendix C (pp. 35–36).

---

### Author Response · Authors · 2026-04-26
**Response to reviewers**

We sincerely thank all reviewers for their careful reading, constructive feedback, and detailed suggestions. We found the comments very helpful for improving both the technical clarity and the empirical support of the paper. In the revised manuscript, all changes are highlighted in red. We provide detailed responses to each comment below.

Thank you for your time and consideration.

Sincerely,
Authors

---

### Decision · Action_Editor_QxYT · 2026-05-20

**Recommendation:** Accept with minor revision

**Additional Comments:**

One point that should still be sharpened concerns the interpretation of balancing. In parts of the literature, balancing is sometimes described as correcting confounding, but this is potentially misleading: balancing is better understood as a finite-sample tool for stabilizing estimation and reducing variance, not as removing unobserved confounding. I recommend clarifying this distinction. This was also the original claim of Shalit et al. and is mirrored in more recent discussions / empirical analyses such as Melnychuk et al. (Bounds on Representation-Induced Confounding Bias for Treatment Effect Estimation; ICLR 2025); Melnychuk et al. (Orthogonal representation learning for estimating causal quantities; AISTATS 2026); Melnychuk et al (Overlap-Adaptive Regularization for Conditional Average Treatment Effect Estimation; ICLR). I guess making the connection to this literature stream on representation learning for causal effect estimation explicit would be helpful!


Minor

- Typo: whewre
- If the authors want, they can shorten a bit the new materials on p4/5

**Audience:**

Yes

**Audience Explanation:**

The paper is relevant to TMLR because it addresses a central problem in causal machine learning: estimating heterogeneous treatment effects from observational data under limited overlap. The main idea—learning representations practically useful and connects representation learning, contrastive learning, and CATE estimation.

**Claims And Evidence:**

Yes

**Claims Explanation:**

The paper proposes HERMES, a Siamese-network framework to estimated treatment responses with a balancing-like approach in a latent space. In the revision, the authors substantially improved the paper by adding ablations, robustness analyses, clearer positioning relative to prognostic score literature, etc. The method has some theoretical intuition but not a fully-fledged theory as one would expect for a regular conference submission.

---

> ### Author Response · Authors · 2026-05-31
> **Camera-ready revision of the paper**
>
> We sincerely appreciate the time and effort that the Action Editor and reviewers have dedicated to evaluating our manuscript. The insightful comments and constructive suggestions provided during the review process have been extremely valuable in helping us improve the quality, clarity, and overall presentation of the paper.
>
> The paper has been revised according to the comments raised in the recommendation. In particular, we clarified the interpretation of balancing throughout the paper, emphasizing that balancing should be understood as a finite-sample tool for improving empirical treated–control comparability, overlap, and stability under standard identification assumptions, rather than as a mechanism for removing unobserved confounding. We also revised the terminology accordingly, added explicit connections to recent work on representation-induced confounding bias, orthogonal representation learning, and overlap-adaptive regularization for CATE estimation, corrected the reported typo, and slightly shortened the newly added material where appropriate. In the following, we address every comment providing a point-to-point response.
>
> COMMENT 1: One point that should still be sharpened concerns the interpretation of balancing. In parts of the literature, balancing is sometimes described as correcting confounding, but this is potentially misleading: balancing is better understood as a finite-sample tool for stabilizing estimation and reducing variance, not as removing unobserved confounding. I recommend clarifying this distinction. This was also the original claim of Shalit et al. and is mirrored in more recent discussions / empirical analyses such as Melnychuk et al. (Bounds on Representation-Induced Confounding Bias for Treatment Effect Estimation; ICLR 2025); Melnychuk et al. (Orthogonal representation learning for estimating causal quantities; AISTATS 2026); Melnychuk et al (Overlap-Adaptive Regularization for Conditional Average Treatment Effect Estimation; ICLR). I guess making the connection to this literature stream on representation learning for causal effect estimation explicit would be helpful!
>
> RESPONSE 1: We thank the editor for this helpful comment. We agree that balancing should not be interpreted as a mechanism that removes unobserved confounding. To address this concern, we revised the manuscript to clarify that balancing is intended as a finite-sample strategy to improve comparability between treated and control units and to support more stable counterfactual estimation under the standard assumptions of causal inference. It does not replace the ignorability assumption and cannot eliminate bias due to unmeasured confounders. Accordingly, we revised the discussion throughout the manuscript to make this distinction more explicit and to avoid any wording that could suggest that the learned representation removes confounding. We now emphasize that representation-learning approaches can only adjust for information contained in the observed covariates and that causal identification still relies on the usual assumptions.
>
> COMMENT 2 (Minor):
> • Typo: Typo: whewre
> • If the authors want, they can shorten a bit the new materials on p4/5
>
> RESPONSE 2: Addressed.